# A New Operational Mediterranean Diurnal Optimally Interpolated SST Product within the Copernicus Marine Service

Andrea Pisano[1], Daniele Ciani[1], Salvatore Marullo[1,2] Rosalia Santoleri[1], Bruno Buongiorno Nardelli[3]

[1]CNR-ISMAR, Via del Fosso del Cavaliere 100, Rome, 00133, Rome, Italy
[2]ENEA, Via Enrico Fermi, 45, 00044 Frascati, Italy
[3]CNR-ISMAR, Calata Porta di Massa, Napoli, 80133, Italy

*Correspondence to*: Andrea Pisano (andrea.pisano@cnr.it)

**Abstract.** Within the Copernicus Marine Service, a new operational MEDiterranean Diurnal Optimally Interpolated Sea Surface Temperature (MED DOISST) product has been developed. This product provides hourly mean maps (Level-4) of sub-skin SST at 1/16° horizontal resolution over the Mediterranean Sea from January 2019 to present. Sub-skin is the temperature at ~1 mm depth of the ocean surface, and then potentially subject to a large diurnal cycle. The product is built by combining hourly SST data from the Spinning Enhanced Visible and InfraRed Imager (SEVIRI) on board Meteosat Second Generation and model analyses from the Mediterranean Forecasting System (MedFS) through optimal interpolation. SEVIRI and MedFS (first layer) SST data are respectively used as the observation source and first-guess. The choice of using a model output as first-guess represents an innovative alternative to the commonly adopted climatologies or previous day analyses, providing physically consistent estimates of hourly SSTs. The accuracy of the MED DOISST product is assessed here by comparison against surface drifting buoy measurements, covering the years 2019 and 2020. The diurnal cycle reconstructed from DOISST is in good agreement with the one observed by independent drifter data, with a mean bias of 0.041 ± 0.001 K and root-mean-square difference (RMSD) of 0.412 ± 0.001 K. The new SST product is more accurate than the input MedFS SST during the central warming hours, when the model, on average, underestimates drifter SST by one tenth of degree. The capability of DOISST to reconstruct diurnal warming events, which may reach intense amplitudes larger than 5 K in the Mediterranean Sea, is also analysed. Specifically, a comparison with the OSTIA diurnal skin SST product, SEVIRI, MedFS and drifter data, shows that the DOISST product is able to reproduce more accurately diurnal warming events larger than 1 K. This product can contribute to improve the prediction capability of numerical models that assimilate or correct the heat fluxes starting from Level-4 SST data, as well as the monitoring of surface heat budget estimates and temperature extremes which can have significant impacts on the marine ecosystem.

The full MED DOISST product (released on 04 May 2021) is available upon free registration at https://doi.org/10.48670/moi-00170 (Pisano et al., 2021). The reduced subset used here for validation and review purposes is openly available at https://doi.org/10.5281/zenodo.5807729 (Pisano, 2021).

## 1 Introduction

In the last decades, the development of accurate satellite-based Sea Surface Temperature (SST) products required an increasing effort to meet an ever-growing request from scientific, operational and emerging policy needs. Indeed, infrared and/or microwave satellite radiometers allow a systematic and synoptic mapping of the ocean surface temperature (under clear-sky conditions for the infrared and in the absence of rain for the microwave bands) with spatial resolutions from one to few kilometers and temporal sampling from hourly to daily (Minnett et al., 2019). This almost continuous coverage represents a unique characteristic of satellite thermal data, which is clearly not achievable with the use of in situ measurements alone. Indeed, though in situ sensors reach significantly higher accuracy than satellite sensors, with uncertainties that can reach $O(10^{-2} K)$, they provide pointwise seawater temperature measurements, generally characterized by a poor and non-uniform sampling of the ocean surface.

There is a huge variety of satellite-based SST datasets, characterized by different nominal resolutions as well as temporal and spatial (global or regional) coverage, and based on different processing algorithms and satellite sensors, but designed to provide highly accurate SST estimates (Yang et al., 2021). Operational datasets are typically distributed in near real time (NRT), delayed-mode or as reprocessed datasets, and may include different processing levels, from single satellite passes processed to provide valid SST values in the original observation geometry, the so-called Level-2 (L2), to images remapped onto a regular grid, also known as Level-3 (L3), up to the spatially complete Level-4 (L4), interpolated over fixed regular grids. These latter are required by several applications since the lower levels are typically affected by several data voids (due to clouds, rain, land, sea-ice, or other environmental factors depending on the type of sensors). The timely availability of SST data, ranging from a few hours to a few days before real time, allows their use as boundary condition and/or assimilation in meteorological and ocean forecasting systems (Waters et al., 2015), to improve the retrieval of ocean surface currents (Bowen et al., 2002; Rio and Santoleri 2018), and monitor some weather extreme events, such as marine heatwaves (Oliver et al., 2021). The reprocessing of long-term SST data records, typically covering the satellite era (1981-present), aims to provide more stable and consistent datasets, complementing the NRT production, to be used to investigate climate variability and monitor changes from interannual to multi-decadal timescales (Deser et al., 2010), including e.g. SST trends' estimates (Good et al., 2007; Pisano et al., 2020). The Copernicus Marine Service is one of the main examples of how satellite observations, including not only SST but a wide range of surface variables (e.g., sea surface salinity, sea surface height, ocean color, winds and waves),

are exploited to derive and disseminate high-level products (Le Traon et al., 2019), namely L4 data, in order to be directly usable for downstream applications.

The majority of the existing L4 SST datasets are provided as daily, weekly or monthly averaged fields (see e.g. Fiedler et al., 2019; Yang et al., 2021). Examples of well-known state-of-the-art SST daily datasets include the Global Ocean Sea Surface Temperature and Sea Ice (OSTIA) dataset (Good et al., 2020), the European Space Agency (ESA) Climate Change Initiative (CCI) Reprocessed Sea Surface Temperature Analyses (Merchant et al., 2019), and the NOAA Daily Optimally Interpolated SST (OISST) v2.1 dataset, previously known/referred to as Reynolds SST analysis (Huang et al., 2021). Though a daily resolution is generally sufficient to meet the requirements of many of the oceanographic applications, it does not resolve the SST diurnal cycle, the typical day-night SST oscillation mainly driven by solar heating. Within the oceanic thermal skin layer (few µm to 1 mm), SST is typically subject to a large potential diurnal cycle (especially under low wind speed and strong solar heating conditions) reaching amplitudes up to 3 K in the world oceans (Gentemann et al., 2008; Gentemann and Minnett, 2008).

The SST diurnal cycle has several implications on mixed layer dynamics, air-sea interaction and the modulation of the lower atmosphere dynamics. The most direct consequence of the SST diurnal amplitude variability is certainly on air-sea fluxes. Clayson and Bogdanoff (2013) estimated that the diurnal SST cycle contributes approximately 5 $Wm^{-2}$ to the global ocean-atmosphere heat budget with peaks of about 10 $Wm^{-2}$ in the Tropics. The inclusion of a realistic diurnal SST cycle in atmospheric numerical simulation also has a non-negligible impact on cloud dynamics. Chen and Houze (1997) have shown that in the Tropical Warm Pool, where extreme localized warming events occur, the diurnal warming can contribute to modulate the evolution of convective clouds and, more in general, can impact the ocean-atmosphere coupling in numerical models, producing a more realistic spatial pattern of warming and precipitation (Bernie et al., 2008). Overall, the diurnal cycle of SST is generally underestimated in current ocean models and the assimilation of SST at high temporal frequency has the potential to improve sea surface variability and mixed layer accuracy (Storto and Oddo, 2019).

In principle, the best opportunity to measure the diurnal cycle comes from infrared radiometers on board geostationary satellites. Their observations are sufficiently accurate and frequent to resolve the diurnal signal variability whenever cloud cover is not too persistent. An example is provided by the Spinning Enhanced Visible Infra-Red Imager (SEVIRI) onboard the Meteosat Second Generation (MSG) geostationary satellite covers. The operational retrieval of SST from MSG/SEVIRI (managed by the European Organization for the Exploitation of Meteorological Satellites, EUMETSAT, Ocean and Sea-Ice Facility, OSI-SAF) produces L3C hourly sub-skin SST products by aggregating 15 minutes (MSG/SEVIRI) observations within 1 hour. The sub-skin SST is the temperature at the base of the conductive laminar sub-layer of the ocean surface, as defined by the Group of High Resolution SST (GHRSST, see e.g. Minnett et al., 2019). In practice, this is the temperature at ~1 mm depth (see e.g., osisaf_cdop3_ss1_pum_msg_sst_data_record.pdf (eumetsat.int)), and thus particularly sensitive to diurnal warming.

For the global ocean, the Operational Sea surface Temperature and sea Ice Analysis (OSTIA) diurnal product (While et al., 2017) provides daily gap-free maps of hourly mean skin SST at 0.25° x 0.25° horizontal nominal resolution, using in situ and satellite data from infrared radiometers. The skin temperature is defined as the temperature of the ocean measured by an infrared radiometer (typically aboard satellites) and represents the temperature of the ocean within the conductive diffusion-dominated sub-layer at a depth of ~10-20 µm (GHRSST, Minnett et al., 2019). This system produces a skin SST by combining the OSTIA foundation SST analysis (Good et al., 2020) with a diurnal warm-layer temperature difference and a cool skin temperature difference derived from numerical models.

At regional scale, a method to reconstruct the hourly SST field over the Mediterranean Sea from SEVIRI data has been proposed by Marullo et al. (2014, 2016). The reconstruction is based on a blending of satellite (SEVIRI) observations and numerical model analyses (used as first-guess) in an optimal interpolation scheme. Model analyses are provided by the Mediterranean Forecasting System, MedFS (Clementi et al., 2021), and distributed through the Copernicus Marine Service (hereafter referred to as Copernicus). Though model analyses by definition also assimilate observations, which could thus in principle include hourly SEVIRI data, in the present configuration, MedFS is not able to deal with such frequent updates and basically only uses one estimation of foundation SST to correct surface fluxes (see section 2.2). As such, the approach presented here represents an effective way to improve the reconstruction of SST daily cycle from high-repetition satellite measurements. Previous works demonstrated the capability of SEVIRI to resolve the SST diurnal variability and to reconstruct accurate L4 SST hourly fields over the Mediterranean Sea, a basin that exhibits large diurnal SST variations (Buongiorno Nardelli et al., 2005; Minnett et al., 2019) that can easily exceed extreme values (~5 K) as observed in the Tropical Pacific (Chen and Houze 1997), in the Atlantic Ocean and other marginal seas (Gentemann et al., 2008; Merchant et al., 2008). The aim of this paper is to describe the operational implementation of a diurnal optimally interpolated SST (DOISST) product for the Mediterranean Sea (MED), building on the algorithm by Marullo et al. (2014, 2016). The DOISST product routinely provides hourly mean maps of sub-skin SST at 1/16° horizontal resolution over the Mediterranean Sea from January 2019 to present. The assessment presented here for the DOISST product covers two complete years (2019-2020), thus extending previous similar validations (Marullo et al., 2016).

## 2 The data

### 2.1 Satellite data

Input satellite SST is derived from the SEVIRI sensor onboard the Meteosat Second Generation (Meteosat-11) satellite. SEVIRI has a repeat cycle of 15 minutes over the 60S-60N and 60W-60E domain: Atlantic Ocean, European Seas and western Indian Ocean. The retrieval of SST from Meteosat-11/SEVIRI is managed by EUMETSAT OSI-SAF, which provides sub-

skin SST data as aggregated (L3C) hourly products remapped onto a 0.05° regular grid. Hourly products result from compositing the best SST measurements available in one hour and are made available in near real time with a timeliness of 3 hours (see the OSI-SAF product user manual, https://osi-saf.eumetsat.int/products/osi-206). File format follows the Data Specification (GDS) version 2 from the Group for High Resolution Sea Surface Temperatures (GHRSST, https://podaac-tools.jpl.nasa.gov/drive/files/OceanTemperature/ghrsst/docs/GDS20r5.pdf). The computation of SST in day and night conditions is based on a nonlinear split window algorithm whose coefficients are determined from brightness temperature simulations on a radiosonde profile database, with an offset coefficient corrected relative to buoy measurements. A correction term derived from simulated brightness temperatures with an atmospheric radiative transfer model is then applied to the multispectral derived SST (OSI-SAF PUM, https://osi-saf.eumetsat.int/lml/doc/osisaf_cdop3_ss1_pum_geo_sst.pdf). L3C data are provided with additional information, including quality level and cloud flags. Such quality flags are provided at pixel level, ranging over a scale of five levels with increasing reliability: 1 (="cloudy"), 2 (="bad"), 3 (="acceptable"), 4 (="good") to 5 (="excellent").

The accuracy of Meteosat-11 SST data has been assessed  through comparison with co-located drifting buoys, for day and night data separately covering the period from February to June 2018 (see the OSI-SAF scientific validation report, https://osi-saf.eumetsat.int/lml/doc/osisaf_cdop2_ss1_geo_sst_val_rep.pdf). The mean bias and standard deviation (derived from the differences between SEVIRI SSTs and drifter measurements over a matchup database) during nighttime have been quantified in -0.1 K and 0.53 K, respectively. During daytime, the bias remains practically unchanged (-0.09 K) and the standard deviation slightly higher (0.56 K). These statistics were derived by selecting SEVIRI SST with quality flags $\geq 3$, and it is shown that the quality of SST improves when choosing higher quality levels. A similar validation procedure (Marullo et al., 2016), but performed over the Mediterranean Sea by using nighttime and daytime data selected with quality flags $\geq 4$, shows that SEVIRI SST bias and standard deviation are -0.03 K and 0.47 K, respectively.

For our purposes, we selected L3C SST data with quality flag $\geq 3$, as also indicated/suggested in the OSI-SAF scientific validation report. A synthesis of the SEVIRI SST characteristics is reported in Table 1.

**2.2 Model data**

The model output fields of surface temperature are derived from the Mediterranean Forecasting System (MedFS), a numerical ocean prediction system that produces analyses, reanalyses and short term forecasts for the Mediterranean Sea and the eastern Atlantic ocean in the 18°W to 6°W - 31°N to 45°N box , to better resolve the exchanges at the Strait of Gibraltar. MedFS is part of the Copernicus Marine Service, and provides regular and systematic information about the physical state of the Mediterranean Sea (https://doi.org/10.25423/CMCC/MEDSEA_ANALYSISFORECAST_PHY_006_013_EAS6; last access: 15 July 2022; Clementi et al., 2021). MedFS is a coupled hydrodynamic-wave model with data assimilation component,with a horizontal grid resolution of 1/24˚ (~4 km) and 141 unevenly spaced vertical levels (Clementi et al., 2017a,b; Pinardi et al., 2003). The Ocean General Circulation Model is based on the Nucleus for European Modelling of the Ocean (NEMO v3.6)

(Oddo et al., 2014, 2009), while the wave component is provided by Wave Watch-III. The model solutions are corrected by a variational data assimilation scheme (3DVAR) of temperature and salinity vertical profiles and along track satellite sea level anomaly observations (Dobricic and Pinardi 2008). The Copernicus Mediterranean SST L4 product (https://doi.org/10.48670/moi-00172; last access: 15 July 2022) is used for the correction of surface heat fluxes with the relaxation constant of 110 $Wm^{-2}K^{-1}$ centered at midnight since the product provides foundation SST (~SST at midnight).

The MedFS product is produced with two different cycles: a daily cycle for the production of forecasts (i.e., ten-days forecast on a daily basis), and a weekly cycle for the production of analyses. For our purposes, only hourly mean SST fields, which correspond to the first vertical level of the model centered at ~1 m from the surface, are selected. The accuracy of SST data has been quantified via a RMSD of $0.57 \pm 0.11$ °C and a bias of $0.14 \pm 0.09$ °C obtained through a comparison with satellite-based L4 SST data (see https://catalogue.marine.copernicus.eu/documents/QUID/CMEMS-MED-QUID-006-013.pdf). A synthesis of the MedFS SST characteristics is reported in Table 1.

**2.3 In situ data**

Surface drifting buoys have been used for validation purposes (Section 4). Since there are no in situ instruments able to routinely measure skin/sub-skin SSTs, the commonly adopted validation procedure is to use drifters' data, also due to their high accuracy and closeness to the sea surface (their representative depth attains around ~20 cm; Reverdin et al., 2010), and to their abundance compared to other in situ instruments, which allows to achieve a more consistent and homogeneous temporal and spatial coverage. Of course, these observations are affected by a representativeness error when compared to sub-skin SSTs, which is typically quantified in terms of a bias between the two estimates.

Drifter data have been obtained from the Copernicus IN SITU (INS) TAC (identified through https://doi.org/10.48670/moi-00044 for the Mediterranean Sea, and https://doi.org/10.48670/moi-00043 for the Northeastern Atlantic ocean; last access: 15 July 2022), which collects and distributes a variety of physical and biogeochemical seawater measurements, provided with the same homogeneous file format . Each in situ measurement, including drifters, undergoes automated quality controls before its distribution. The quality of the data is expressed by control flags indexed from 0 to 9, with the value of 1 indicating best quality. Drifter data have been used to compile an hourly matchup database (section 4.1) over which validation statistics have been produced(section 4.2). A synthesis of the drifter SST characteristics is reported in Table 1.

**2.4 OSTIA diurnal**

The OSTIA diurnal skin SST product (While et al., 2017) provides gap-free global maps of hourly mean skin SST at 0.25° x 0.25° horizontal resolution, obtained by combining in situ and infrared satellite data. This product is operationally produced by the Met Office within the Copernicus Marine Service (https://doi.org/10.48670/moi-00167; last access: 15 July 2022), and created using the Operational Sea surface Temperature and Ice Analysis (OSTIA) system (Good et al., 2020). The OSTIA

system also produces a global daily average foundation SST L4 product (https://doi.org/10.48670/moi-00165; last access: 15 July 2022). Since the skin SST can be considered as the sum of three components, namely the foundation SST, the warm layer and the cool skin, the OSTIA diurnal product is created by adjusting the OSTIA foundation SST analysis with a modelled diurnal warm layer analysis (which assimilates satellite observations) and a cool skin model, based respectively on the Takaya (Takaya et al., 2010) and Artale models (Artale et al., 2002). Assimilation into the warm layer model makes use of SEVIRI, GOES-W and MTSAT-2 geostationary infrared sensors, and of the polar orbiting VIIRS radiometer. Further details on the method can also be found in Copernicus PUM (https://catalogue.marine.copernicus.eu/documents/PUM/CMEMS-SST-PUM-010-014.pdf)). A synthesis of the OSTIA diurnal SST characteristics is reported in Table 1.

| SST | | | | | | | |
|------|------------|----------------|-------------|----------|--------------|-------------|------------|
| Source | Definition | Vertical level | Spatial res. | Temporal res. | Spatial coverage | Temporal coverage | Processing level |
| MedFS | Depth SST | 1 m (first model layer) | 0.042°x0.042° | Hourly | 17.3°W–36.3°E, 30.2°N–46°N | 2019-Present | Model output |
| SEVIRI | Sub-skin SST | ~1 mm (surface only) | 0.05°x0.05° | Hourly | 60°W–60°E, 60°S–60°N | 2015-Present | L3C |
| OSTIA diurnal | Skin SST | ~10-20 µm (surface only) | 0.25°x0.25° | Hourly | Global | 2015-Present | L4 |
| Surface Drifting Buoys | Depth SST | ~20 cm (surface only) | Not applicable | Hourly | 30°W–36.5°E, 20°N–55°N | 2010-Present | L2 |

**Table 1.** Summary of the SST products used to produce (MedFS and SEVIRI), validate (surface drifting buoys), and intercompare (all) the DOISST product. The SST nomenclature (skin, sub-skin, and depth) follows the Group for High Resolution Sea Surface Temperatures (GHRSST) definitions (https://podaac-tools.jpl.nasa.gov/drive/files/OceanTemperature/ghrsst/docs/GDS20r5.pdf).

## 3 The Mediterranean diurnal optimally interpolated SST product

### 3.1 Product overview

The Mediterranean diurnal optimally interpolated SST (hereafter referred to as MED DOISST) operational product consists of hourly mean gap-free (L4) satellite-based estimates of the sub-skin SST over the Mediterranean Sea (plus the adjacent Eastern Atlantic box, see Section 2.2) at 0.0625° x 0.0625° grid resolution, from 1st January 2019 to near real time. Specifically, the product is updated daily and provides 24 hourly mean data of the previous day, centered at 00:00, 01:00,

02:00,…,23:00 UTC. The MED DOISST product is published on the Copernicus on line catalogue and identified as
SST_MED_PHY_SUBSKIN_L4_NRT_010_036 (product reference) and cmems_obs-sst_med_phy-sst_nrt_diurnal-oi-
0.0625deg_PT1H-m (dataset reference). Further details on the product characteristics are provided in Table 2.
DOISST is the result of a blending of SEVIRI sub-skin SSTs and MedFS SSTs (as detailed in section 3.2), the former
representative of a depth of 1 mm and the latter of 1 m. Then, the DOISST effective depth does, in principle, vary between 1
mm up to 1 m, depending on how the relative amount of satellite observations used in the interpolation. However, diurnal
warming is significantly reduced under cloudy conditions (when SEVIRI data are not available), so that, in those cases, the
difference between the SST at 1 m and the sub-skin SST is small. Under clear sky conditions, SEVIRI observations will
dominate the retrieved SST, so the DOISST product can be safely defined as representative of sub-skin values.

**Copernicus Marine Service Product ID:** SST_MED_PHY_SUBSKIN_L4_NRT_010_036

| Dataset ID: cmems_obs-sst_med_phy-sst_nrt_diurnal-oi-0.0625deg_PT1H-m | |
|---|---|
| **General description** | The Copernicus Mediterranean diurnal product provides near-real-time, hourly mean, gap-free (L4) sub-skin SST fields over the Mediterranean Sea and the adjacent Atlantic box over a 0.0625°x0.0625° regular grid, covering the period from 2019 to present (one day before real time). This product is built from optimal interpolating the Level-3C (merged single-sensor, L3C) SEVIRI data as observations and the Copernicus Mediterranean MedFS analyses as first-guess. 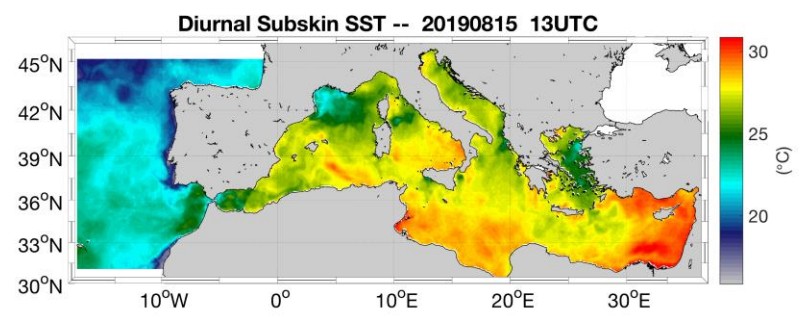 |
| **Horizontal resolution** | 0.0625° x 0.0625° (1/16°) degrees [871x253] |
| **Temporal resolution** | Hourly |
| **Spatial coverage** | Mediterranean Sea + adjacent North Atlantic box (W=-18.1250, E=36.2500, S=30.2500, N=46.0000) |
| **Temporal coverage** | 2019/01/01 – near real time (-14H) |
| **Vertical level** | ~1 mm (surface only) |
| **Variables** | Sub-skin SST (K) <br> Analysis Error (%) |
| **Format** | NetCDF – CF-1.4 convention compliant |
| **DOI** | https://doi.org/10.48670/moi-00170 |

| Comments | Eventual updates of this product will be described in the corresponding Product User Manual (PUM) and Quality Information Document (QUID) available on the Copernicus Marine Service on line catalogue. |
|---|---|


**Table 2.** The Copernicus Marine Service MED DOISST product description synthesis.

**3.2 Background**
The reconstruction of gap-free hourly mean SST fields is based on a blending of SEVIRI (satellite) observations and MedFS
(model) analyses (used as first-guess/background) using optimal interpolation (OI), following the approach proposed by
Marullo et al. (2014). The OI method determines the optimal solution to the interpolation of a spatially and temporally variable
field with data voids, where "optimal" is intended in a least square sense (see e.g. Bretherton et al., 1976). The optimally
interpolated variable, or analysis ($F_a$), is obtained as follows:

$$F_a(x,t) = F_b(x,t) + \sum_{i,j=1}^{n} W_{i,j}( F_{obs,i}(x,t) - F_b(x,t)) \qquad (1)$$


In practice, the analysis $F_a(x,t)$ at a particular location in space and time $(x,t)$ is obtained as a correction to a background
field ($F_b(x,t)$). The correction is estimated as a linear combination of the observation anomalies ($F_{obs} - F_b$), where the
coefficients $W_{i,j}$ are obtained by minimizing the analysis error variance.
The choice of using MedFS SST as first-guess represents the best alternative to the use of climatologies or previous day
analyses, as usually done by other schemes to produce daily SST L4 maps, since the model provides physically consistent
estimates of hourly SSTs (Marullo et al., 2014). In fact, the model takes into account the effect of air-sea interactions by
imposing external forcings that drive momentum and heat exchanges at the upper boundary. As such, it is able to reproduce at
least part of the diurnal warming effects, that are driven by the forcing diagnosed from atmospheric model analyses. Using
MedFS SST as a first-guess means we are treating the hourly satellite data as corrections to the hourly model data. The
observation anomalies are generally small and mostly drive corrections to the spatial patterns, while displaying a reduced
diurnal cycle. Anomaly data from different times of the day can thus be more "safely" used to build the interpolated field at
each reference time (with different weights). Unfortunately, the first MedFS model layer is at 1 m depth, which means that it
will generally underestimate the diurnal cycle anyway. While 1D models could in principle be used to better reproduce sub-
skin SST from model data, the approach presented here is focusing on providing estimates that are as close as possible to the
original satellite data, avoiding the complications of setting up an additional preprocessing step just to improve the first-guess.

**3.3 Processing chain**
The DOISST system ingests merged single-sensor (L3C) SEVIRI SST as the observation source, and MedFS SST (first layer)
as first-guess.
The data sub-sampling strategy, inversion technique and numerical implementation of the optimal interpolation scheme are
based on the Copernicus NRT MED SST processing chain (Buongiorno Nardelli et al., 2013), which provides daily mean
fields of foundation SST over the Mediterranean Sea (https://doi.org/10.48670/moi-00172; last access: 15 July 2022). Here,
the diurnal SST chain is organized in three main modules (Fig. 1).

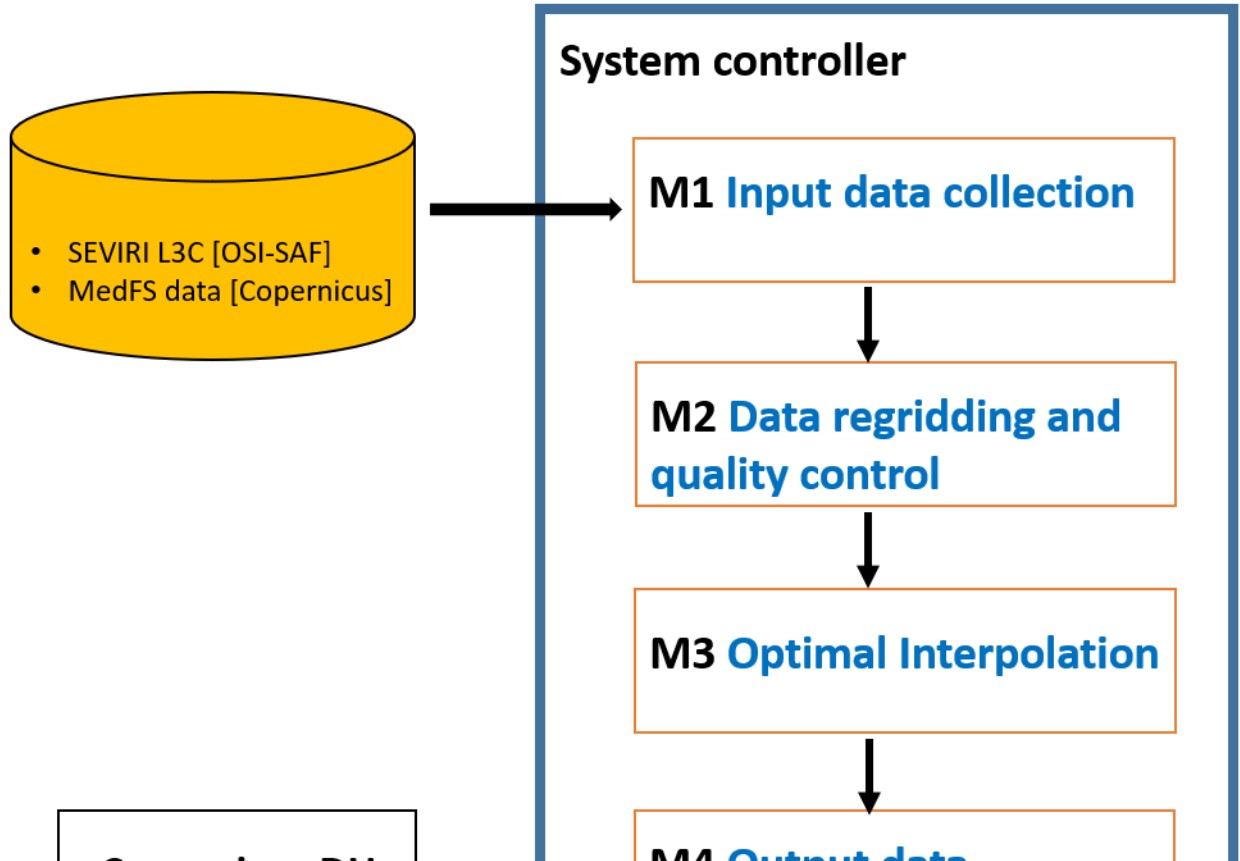

**Figure 1.** Schematic diagram of the processing chain used for the MED DOISST SST product.

Module M1 manages the external interfaces to get both upstream L3C and model data: hourly mean L3C sub-skin SST data at
0.05° grid resolution are downloaded from OSI-SAF while hourly MedFS SST data at 1.0182 meter (first level) at 0.042° grid
resolution from the Copernicus Marine Service.
Module M2 extracts and regrids (through bilinear interpolation) both SEVIRI L3C and MedFS SST data over the DOISST
geographical domain at 1/16° grid resolution (see Table 2). A selection over SEVIRI is performed by flagging the pixels with
quality flag < 3.
Module M3 performs a space-time optimal interpolation (OI) algorithm. L4 data are obtained as a linear combination of the
SST anomalies, weighted directly with their correlation to the interpolation point and inversely with their cross-correlation and
error (Eq. 1). Correlations are typically expressed through analytical functions with predefined spatial and temporal de-
correlation lengths. Here, the covariance function $f(r, \Delta t)$ is the one defined in Marullo et al. (2014), and given as the product
of a spatial and temporal component:
$$f(r, \Delta t) = \left[ \alpha . e^{-\frac{r}{R}} + \frac{1-\alpha}{(1+r)^c} \right] . e^{-\left(\frac{\Delta t}{T}\right)^d} \qquad (2)$$


where r is the distance (in km) between the observation and the interpolation point; Δt is the temporal difference (in hours)
between the observation and the interpolation point; R = 200 km is the decorrelation spatial length; T = 36 h is the decorrelation
time length; the other parameters are set as follows: a = 0.70, c = 0.26, d = 0.4. All these parameters have been derived in
Marullo et al. (2014), deduced from a nonlinear least square fit between the estimated temporal and spatial correlations In
practice, the weights in expression (1) are computed directly from the analytical function (2).
The input data are selected only within a limited sub-domain (within a given space-time interval, also called "influential"
radius), with a temporal window of ±24 h (this the result of several trials over a large variety of environmental conditions;
Marullo et al., 2014) and a spatial search radius of about 700 km (Buongiorno Nardelli et al., 2013). A check to avoid data
propagation across land is performed between each pixel within the sub-domain and the given interpolation point (eventually
discarded if there are land pixels between the straight line connecting the two points).
The interpolation error (analysis_error field in the L4 file, Table 2) is obtained from the formal definition of the error variance
derived from optimal interpolation theory (e.g., Bretherton et al., 1976). This error ranges between 0-100%, meaning that the
error is almost zero when an optimal number of observations is present within the space-time influential radius, while only
first-guess data are used (i.e. no observations are found within the search radius) when the error is 100%.
The optimal interpolation algorithm is synthetized as follows. For clarity, in order to interpolate an SST map on a given day
at 12:00 UTC the following steps have to be done:
• Download of ±24 hourly SEVIRI L3C and MedFS (first layer) SST fields (in their native spatial resolution) centered
with respect to the interpolation time;
• Extract and regrid over the DOISST geographical domain at 1/16°;
• Retain only SEVIRI data with quality flag ≥ 3;
• Subctract hourly MedFS SSTs from valid SEVIRI SSTs to produce SST anomalies;
• Use SST anomalies as data input for the optimal interpolation analysis;
• Collect anomalies in a space/time window of 700 km/ ±24 h with respect to the interpolation position/time;
• Run Optimal interpolation using the covariance function defined above;
• Add the hourly (at 12:00 UTC) MedFS SST field to the optimally interpolated output again.

Obviously, the symmetric temporal window (±24 hourly) can be applied only for reprocessing. During near-real-time DOISST
processing, the input data are collected starting from 24 h before the interpolation time up to the last available SEVIRI hourly
SST field.
Finally, the main difference with the original method is that all the input observations are interpolated, while in Marullo et al.
(2014) valid SST observations are left unchanged (not interpolated).
**4 Validation of diurnal product**
**4.1 Validation framework**
The accuracy of the MED DOISST product has been assessed through comparison with independent co-located (in space and
time) surface drifting buoy data (matchups). The relative and absolute validation framework is thus based on the compilation
of a matchup database between DOISST, SEVIRI L3C, MedFS (all available at 1/16° as described in section 3.3), and OSTIA
diurnal (kept at its original ¼° resolution), and drifters measurements covering the full years 2019 and 2020. The large number
of drifters provides a rather homogeneous and continuous spatial and temporal coverage over the whole period (Fig. 2) allowing
a robust statistical approach.
Firstly, a pre-selection of high-quality drifter data is performed, retaining only temperatures with quality flag equal to 1 (good)
or 2 (probably good) (see section 2.3). Then, the co-location is carried out on hourly basis, building a matchup database by
collecting the closest (nearest neighbour) SST grid point to the in situ measurement within a symmetric temporal window of
30 minutes with respect to the beginning of each hour. A final quality outlier detection check is carried out by identifying
drifter data for which the module of the difference with respect to satellite observations exceeds n-times the standard deviation
$\sigma$ of the distribution of the differences ($\delta$). At each step n decreases, and data that fall out of the interval $I = [mean(\delta) - n \cdot$
$\sigma, mean(\delta) + n \cdot \sigma]$ are flagged as outliers and removed. For each n, the selected outliers are eliminated and the process is
repeated for the same value of n until no more outliers are detected. Then the system moves to n-1.  The process starts for n=10
and stops at n=3, and removes ~1% of the total original sampling (as expected from a gaussian distribution) of drifter data that
clearly revealed anomalous temperature values.
The main validation statistics are quantified in terms of mean bias and Root-Mean-Square Difference (RMSD) from matchup
temperature differences (namely, SST minus drifter). Each statistical parameter is associated with a 95% confidence interval
computed through a bootstrap procedure (Efron 1994).


### 4.2 Comparison with drifters


### 4.2.1 The mean diurnal cycle


The spatial distribution of DOISST and drifter matchups over the 2019-2020 period, along with their pointwise difference (i.e.,
DOISST minus drifter measurement) shows a rather homogeneous coverage over the most of the DOISST geographical
domain (Fig. 2), although some areas are characterized by quite low coverage, such as the North Adriatic Sea or North Aegean
Sea. The spatial distribution also evidences the predominance of a positive bias, indicating that DOISSTs are warmer than
drifters' temperatures on average.

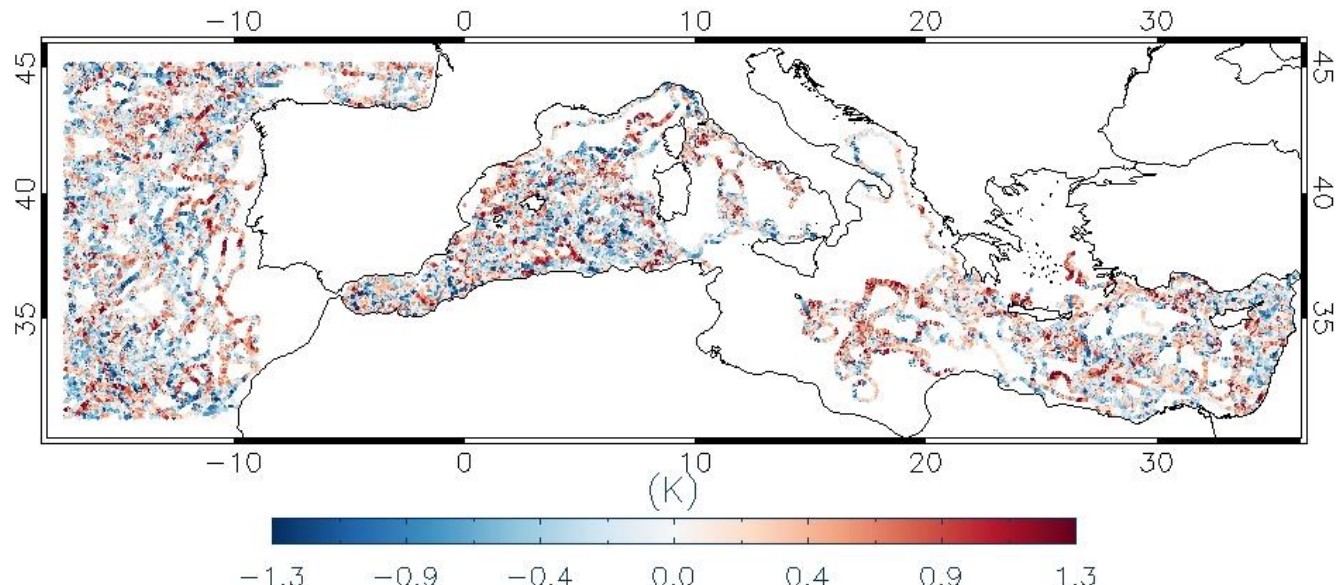


 **Figure 2**. Spatial distribution of the matchup points along with their punctual bias (i.e., SST minus drifter data, K) over the
DOISST geographical domain from 2019/01/01 to 2020/12/31.


The DOISST product shows effectively an overall small positive mean bias of 0.041 ± 0.001 K and a RMSD of 0.412 ± 0.001
K (Table 2). A negative bias of -0.100 ± 0.001 K and slightly larger RMSD of 0.467 ± 0.001 K characterize MedFS SSTs.
Both DOISST and MedFS show high and comparable correlation coefficients (more than 0.99).

| | Period | Mean bias (K) | RMSD (K) | Correlation coeff. | Matchups |
|---|---|---|---|---|---|
| DOISST | 2019-01-01 to 2020-12-31 | 0.041 ± 0.001 | 0.412 ± 0.001 | 0.992 | 548959 |
| MedFS | 2019-01-01 to 2020-12-31 | -0.100 ± 0.001 | 0.467 ± 0.001 | 0.991 | 548959 |

**Table 3.** Summary statistics of DOISST and MedFS SST. Mean bias (K), RMSD (K), and correlation coefficient are derived
from temperature differences against drifters' data over the period 2019-2020. Each statistical parameter is associated with a
95% confidence interval computed through a bootstrap procedure (Efron 1994).

The hourly mean bias of DOISST and MedFS shows similar but opposite behaviour (Fig. 3a, and Table 4). In both cases, the
bias clearly exhibits a diurnal oscillation during the 24 hours but, while the bias of DOISST increases positively during the
central diurnal warming hours, the one of MedFS increases negatively. The DOISST mean bias is practically null between
17:00 to 06:00 local time, ranging between -0.001 and 0.03 K, and highest (~0.1 K) between 10:00 and 13:00 local time. The
MedFS bias oscillates around ~-0.07 K between 23:00 and 07:00 local time. Then, it increases (in absolute value) reaching the
peak of ~-0.16 K between 11:00 and 14:00 and decreases successively. Similar results are obtained for the RMSD, which
increases with diurnal warming (Fig. 3b, Table 4). However, the RMSD of DOISST is less impacted by diurnal variations,
characterized by an amplitude of ~0.04 K against ~0.14 K of MedFS.










(a)                                                    (b)

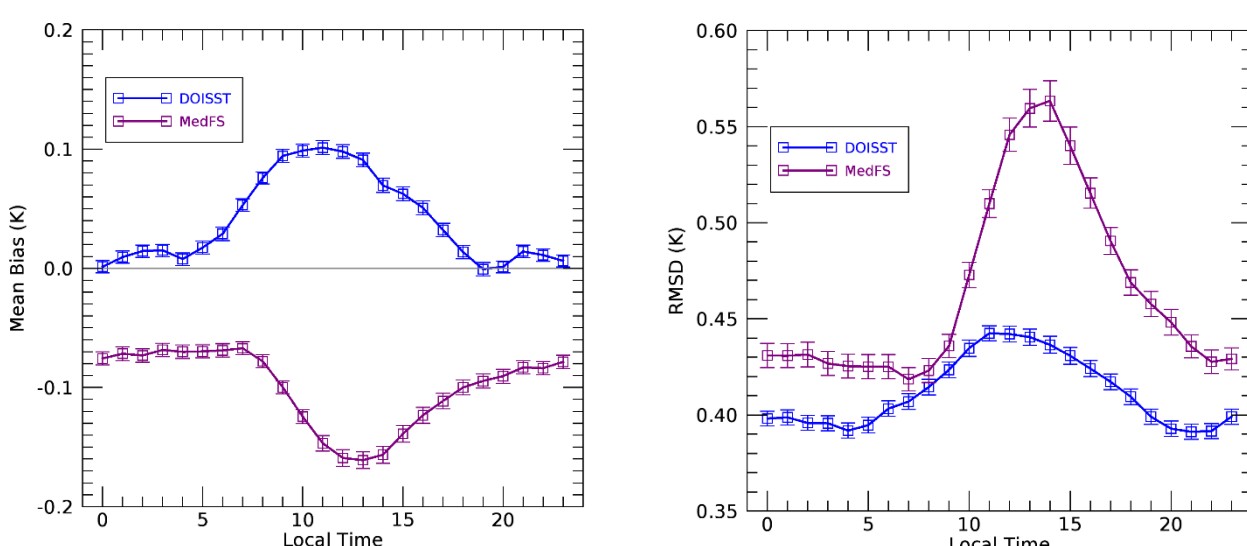

**Figure 3.** (a) Mean bias (K) and (b) RMSD (K) relative to MED DOISST (blue line) and MedFS (purple line) based on the
differences against drifters' data. Mean bias and RMSD are given as hourly mean over the period 2019-2020.

| Hour (local time) | Mean BIAS (K) (DOISST) | RMSD (K) (DOISST) | BUOY-AVAIL | Mean BIAS (K) (MedFS) | RMSD (K) (MedFS) |
|---|---|---|---|---|---|
| HH: 00 | 0.001 ± 0.005 | 0.398 ± 0.004 | 22807 | -0.076 ± 0.006 | 0.431 ± 0.006 |
| HH: 01 | 0.009 ± 0.005 | 0.399 ± 0.004 | 23004 | -0.072 ± 0.006 | 0.431 ± 0.006 |
| HH: 02 | 0.014 ± 0.005 | 0.396 ± 0.004 | 22798 | -0.073 ± 0.005 | 0.431 ± 0.006 |
| HH: 03 | 0.015 ± 0.005 | 0.396 ± 0.004 | 23078 | -0.068 ± 0.006 | 0.427 ± 0.006 |
| HH: 04 | 0.008 ± 0.005 | 0.392 ± 0.004 | 22857 | -0.070 ± 0.005 | 0.425 ± 0.006 |
| HH: 05 | 0.017 ± 0.005 | 0.395 ± 0.004 | 22806 | -0.070 ± 0.005 | 0.425 ± 0.006 |
| HH: 06 | 0.029 ± 0.005 | 0.403 ± 0.004 | 22819 | -0.069 ± 0.006 | 0.425 ± 0.006 |
| HH: 07 | 0.053 ± 0.005 | 0.407 ± 0.004 | 23379 | -0.067 ± 0.005 | 0.419 ± 0.006 |
| HH: 08 | 0.076 ± 0.005 | 0.415 ± 0.004 | 23501 | -0.078 ± 0.006 | 0.423 ± 0.006 |
| HH: 09 | 0.094 ± 0.005 | 0.423 ± 0.004 | 23481 | -0.100 ± 0.006 | 0.436 ± 0.006 |
| HH: 10 | 0.099 ± 0.006 | 0.435 ± 0.004 | 23270 | -0.125 ± 0.006 | 0.473 ± 0.007 |
| HH: 11 | 0.101 ± 0.006 | 0.442 ± 0.004 | 23311 | -0.147 ± 0.006 | 0.510 ± 0.007 |
| HH: 12 | 0.098 ± 0.006 | 0.442 ± 0.004 | 23129 | -0.159 ± 0.007 | 0.546 ± 0.009 |
| HH: 13 | 0.091 ± 0.006 | 0.440 ± 0.005 | 22836 | -0.161 ± 0.007 | 0.560 ± 0.009 |
| HH: 14 | 0.070 ± 0.006 | 0.436 ± 0.004 | 22673 | -0.157 ± 0.007 | 0.563 ± 0.011 |
| HH: 15 | 0.062 ± 0.006 | 0.431 ± 0.004 | 22418 | -0.139 ± 0.007 | 0.540 ± 0.009 |
| HH: 16 | 0.051 ± 0.006 | 0.424 ± 0.004 | 22368 | -0.123 ± 0.007 | 0.515 ± 0.008 |
| HH: 17 | 0.032 ± 0.006 | 0.417 ± 0.004 | 22019 | -0.111 ± 0.006 | 0.491 ± 0.007 |
| HH: 18 | 0.014 ± 0.006 | 0.410 ± 0.004 | 21916 | -0.100 ± 0.006 | 0.469 ± 0.007 |
| HH: 19 | -0.001 ± 0.005 | 0.399 ± 0.004 | 22117 | -0.095 ± 0.006 | 0.458 ± 0.007 |
| HH: 20 | 0.001 ± 0.005 | 0.393 ± 0.004 | 22458 | -0.090 ± 0.006 | 0.448 ± 0.006 |
| HH: 21 | 0.014 ± 0.005 | 0.391 ± 0.004 | 23229 | -0.083 ± 0.005 | 0.436 ± 0.006 |
| HH: 22 | 0.011 ± 0.005 | 0.392 ± 0.004 | 23272 | -0.084 ± 0.006 | 0.428 ± 0.006 |
| HH: 23 | 0.006 ± 0.005 | 0.399 ± 0.004 | 23413 | -0.078 ± 0.006 | 0.429 ± 0.006 |


**Table 4**. Summary statistics of MED DOISST and MedFS products based on the differences against drifters' data over the
matchup points. Mean bias (K), RMSD (K) and number of matchups are given as hourly mean over the period 2019-2020.
Each statistical parameter is associated with a 95% confidence interval computed through a bootstrap procedure (Efron 1994).

The mean diurnal cycle of DOISST (namely, the 24-hour mean SSTs estimated over the matchup dataset) is in very good
agreement, within the error confidence interval, with the SST cycle reconstructed from drifters (Fig. 4). The two diurnal cycles
are practically unbiased between 17:00 and 06:00, while they are biased by ~0.1 K between sunrise and 16:00, coherently with
the DOISST bias oscillation (Fig. 3a). This bias could be related to skin SST warming faster than the temperature at 20 cm
depth. The diurnal cycle of MedFS SST maintains always below that of in situ temperatures, evidencing larger differences
during the central diurnal warming hours (Fig. 4). However, apart from the biases likely induced by the different depths, the
SST amplitude as estimated from the DOISST and MedFS is ~2.3% larger and ~16% smaller than that of drifters, respectively,
suggesting that the model tends to underestimate diurnal variations.

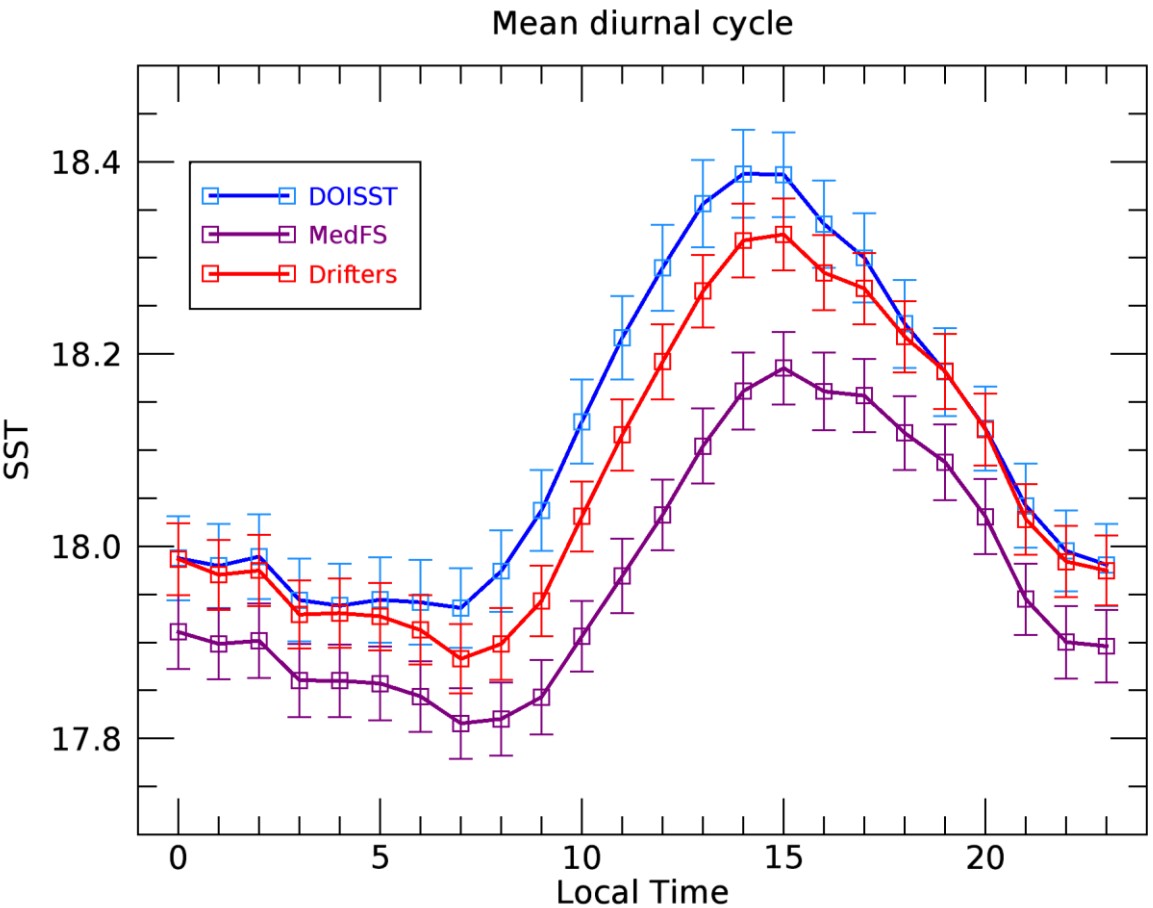

**Figure 4.** Mean diurnal cycle for MED DOISST (blue line), MedFS (purple line) and drifters (red line) computed over the
matchups from 2019 to 2020.

A delay of ~1 hour of MedFS with respect to DOISST and in situ on the onset of diurnal warming and in reaching the maximum
is also evident. This delay could be explained as the physical result of delayed solar heating of the skin layer sensed by the
satellite and of the first model layer. This may also be a consequence of the different packaging of the SEVIRI and MedFS
SST data into the hourly files: MedFS ones are centered at half of every hour (e.g., 12:30), while SEVIRI L3C at the beginning
of each hour (e.g., 12:00) and obtained from collating data within one hour (from 11.30 to 12:29).
The capability of DOISST to capture and realistically reproduce diurnal variability is further investigated by analysing the
seasonally averaged SST diurnal cycle (Fig. 5), computed as for the mean diurnal cycle (by using the matchup dataset) but
over seasons: winter (December to February), spring (March to May), summer (June to August) and autumn (September to
November). The effect of warming in the diurnal SST excursion is clearly more pronounced during spring and summer than
winter and autumn, and reconstructed well in DOISST. During the warmer seasons, the DOISST shows the lower biases (Table
5), estimated in 0.036 ± 0.001 K (spring) and 0.012 ± 0.003 (summer). Conversely, MedFS reaches its higher biases, namely
-0.101 ± 0.001 K (spring) and -0.117 ± 0.003 K (summer). The good agreement between DOISST and drifters during winter
and autumn (Table 5) reveals that the hourly DOISST fields are reconstructed accurately also under cloudy conditions, which
are more frequent during these seasons (Kotsias and Lolis, 2018).

(a)                                                                                       (b)

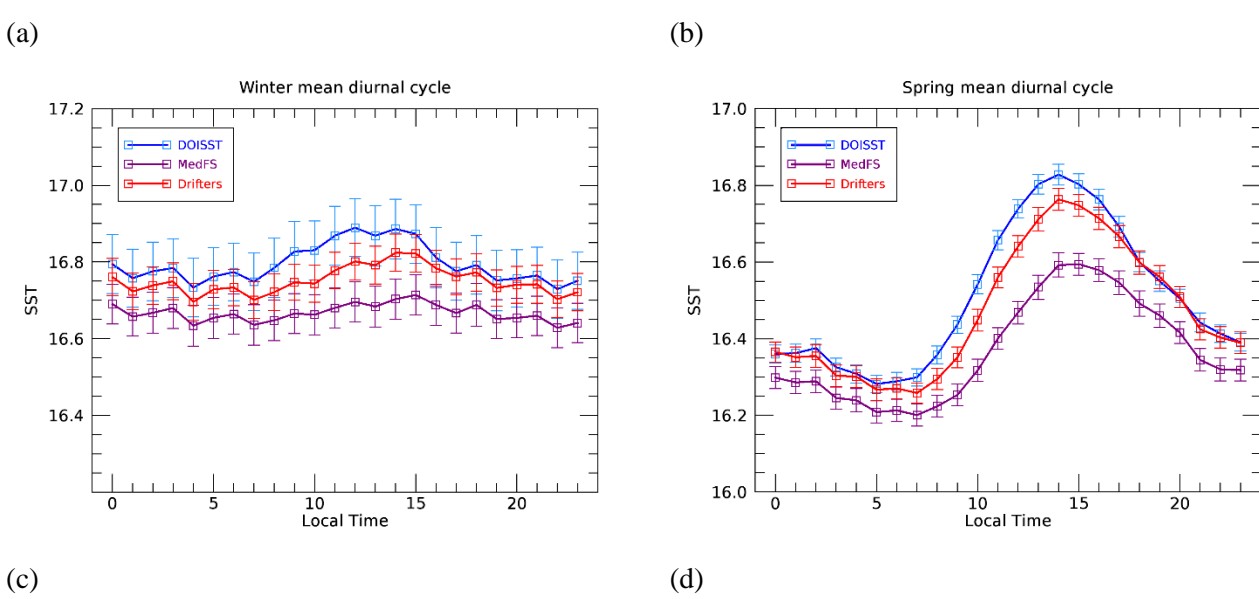

(c)                                                                                       (d)

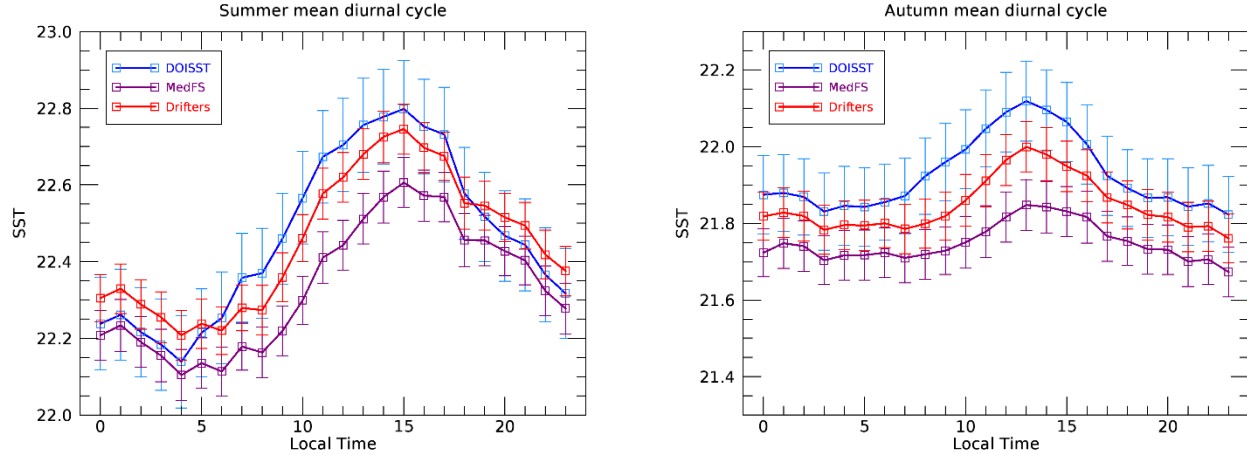

**Figure 5.** Seasonal mean diurnal cycle over the period 2019-2020 for MED DOISST (blue line), MedFS (purple line) and in
situ (red line). (a) Winter (December to February); (b) Spring (March to May); (c) Summer (June to August); and (d) Autumn
(September to November).

| | Period | Mean bias (K) | RMSD (K) | Matchups |
|---|---|---|---|---|
| Winter | DOISST | 0.045 ± 0.003 | 0.428 ± 0.002 | 90247 |
| | MedFS | -0.084 ± 0.004 | 0.563 ± 0.003 | |
| Spring | DOISST | 0.036 ± 0.001 | 0.383 ± 0.001 | 308448 |
| | MedFS | -0.101 ± 0.001 | 0.389 ± 0.002 | |
| Summer | DOISST | 0.012 ± 0.003 | 0.483 ± 0.002 | 74107 |
| | MedFS | -0.117 ± 0.003 | 0.486 ± 0.004 | |
| Autumn | DOISST | 0.079 ± 0.003 | 0.429 ± 0.002 | 76157 |
| | MedFS | -0.098 ± 0.004 | 0.590 ± 0.004 | |

**Table 5.** Summary statistics of DOISST and MedFS SSTs. Mean bias (K) and RMSD (K) are derived from temperature
differences against drifters' data during winter (D-J-F), spring (M-A-M), summer (J-J-A) and autumn (S-O-N) over the period
2019-2020. Each statistical parameter is associated with a 95% confidence interval computed through a bootstrap procedure
(Efron 1994).

### 4.2.2 Diurnal warming events

Diurnal warming (DW) can be defined as the difference between the SST at a given time of the day and the foundation SST (see e.g. Minnett et al., 2019), i.e. the water temperature at a depth such that the daily variability induced by the solar irradiance is negligible. In many cases, the foundation SST coincides with the night minimum SST, namely the temperature that is recorded just before sunrise.

The capability of DOISST to describe diurnal warming events is analysed here in comparison with SEVIRI L3C, OSTIA diurnal, MedFS and drifter data. The evaluation is carried out by computing daily Diurnal Warming Amplitudes (DWAs) from drifters and building a matchup dataset of DWAs as estimated from DOISST, SEVIRI L3C, OSTIA and MedFS data. The inclusion of SEVIRI data is mainly aimed at evaluating the impact of optimal interpolation on the input SEVIRI SSTs, while OSTIA diurnal is used as an intercomparison product. The DWA is estimated here as a difference between the maximum occurred during daytime (10:00-18:00 local time) and the minimum during nighttime (00:00-06:00 local time) (see also Takaya et al., 2010; While et al., 2017). Explicitly, for each day (from 2019 to 2020) and for each drifter the two positions and times relative to the minimum and maximum temperature are stored; over the same times and nearest positions, the temperatures of the other datasets are stored too. The grid resolution of OSTIA diurnal (namely, 0.25° deg.) has been left unchanged since what is needed is just the SST value at a given position, the nearest to the drifter's one.

The scatter plots of DOISST, SEVIRI, OSTIA, and MedFS vs in situ-measured DWA have been computed for the years 2019-2020 (Fig. 6) and organized during spring-summer and winter-autumn seasons (Fig. 7). This choice is aimed at comparing the behaviour of the four products as a function of the seasons, since larger DWA intensities are expected in the spring-summer period.

Overall, there is a good agreement between DOISST and drifter DWAs (Fig. 6a) as confirmed by an almost null mean bias (-0.02 K), low RMSD (0.38 K) and high correlation coefficient (0.82). The largest DW amplitudes reach values as high as 4 K in both DOISST and drifter data. SEVIRI (Fig. 6b) shows the same bias (-0.02 K) of DOISST in reconstructing DWAs but higher RMSD (0.49 K) and lower correlation (0.74). It is relevant to note that the spread of SEVIRI DWAs around the line of perfect agreement is reduced in DOISST, which coherently has a lower RMSD. MedFS (Fig. 6c) clearly underestimates diurnal amplitudes larger than 1 K, and it is characterized by a high mean bias (-0.23 K) and RMSD (0.55 K), and lowest correlation coefficient (0.66). Similarly, OSTIA diurnal (Fig. 6d) underestimates DWAs larger than 1 K, and it is characterized by the highest mean bias (-0.28 K), RMSD of 0.54 K but shows less dispersion than MedFS around the line of perfect agreement (correlation of 0.72).

The majority of DWA events lie between 0-1 K all over the year, but higher values are effectively reached during spring and summer (Fig. 7). During these seasons, it appears more evident the capability of DOISST to better describe DWAs larger than 1 K (mean bias = -0.04 K; RMSD = 0.42 K; corr. = 0.83) compared to SEVIRI (mean bias = -0.05 K; RMSD = 0.53 K; corr. = 0.76) and especially to MedFS (mean bias = -0.27 K; RMSD = 0.65 K; corr. = 0.63) and OSTIA diurnal (mean bias = -0.39

K; RMSD = 0.66 K; corr. = 0.71). During winter and autumn, the overall statistics of the four products get better, clearly due
to the fact that the majority of DWA events range between 0-0.5 K. However, DWA events exceeding 1 K are also observed,
and such intense amplitudes are not found in the model-derived and OSTIA DWAs. Additionally, the good agreement between
DOISST and drifters still confirms that interpolated data do not suffer from the increased cloud cover during winter and autumn
periods.

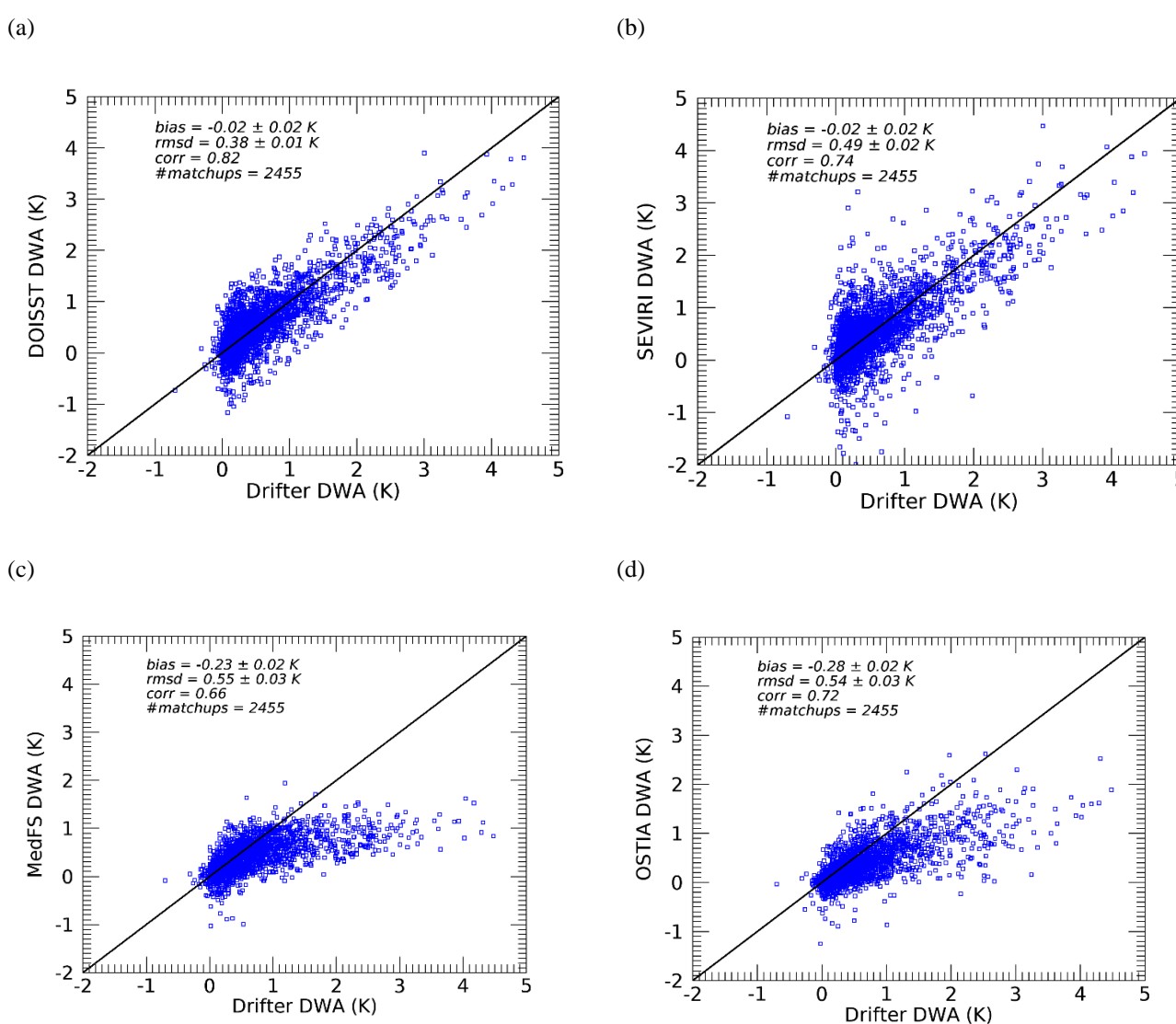

**Figure 6**. DWA scatter plots for (a) DOISST, (b) SEVIRI L3C, (c) MedFS and (d) OSTIA diurnal vs drifters over the period
445  2019-2020.



(a)

(b)

(c)

(d)

(e)

(f)

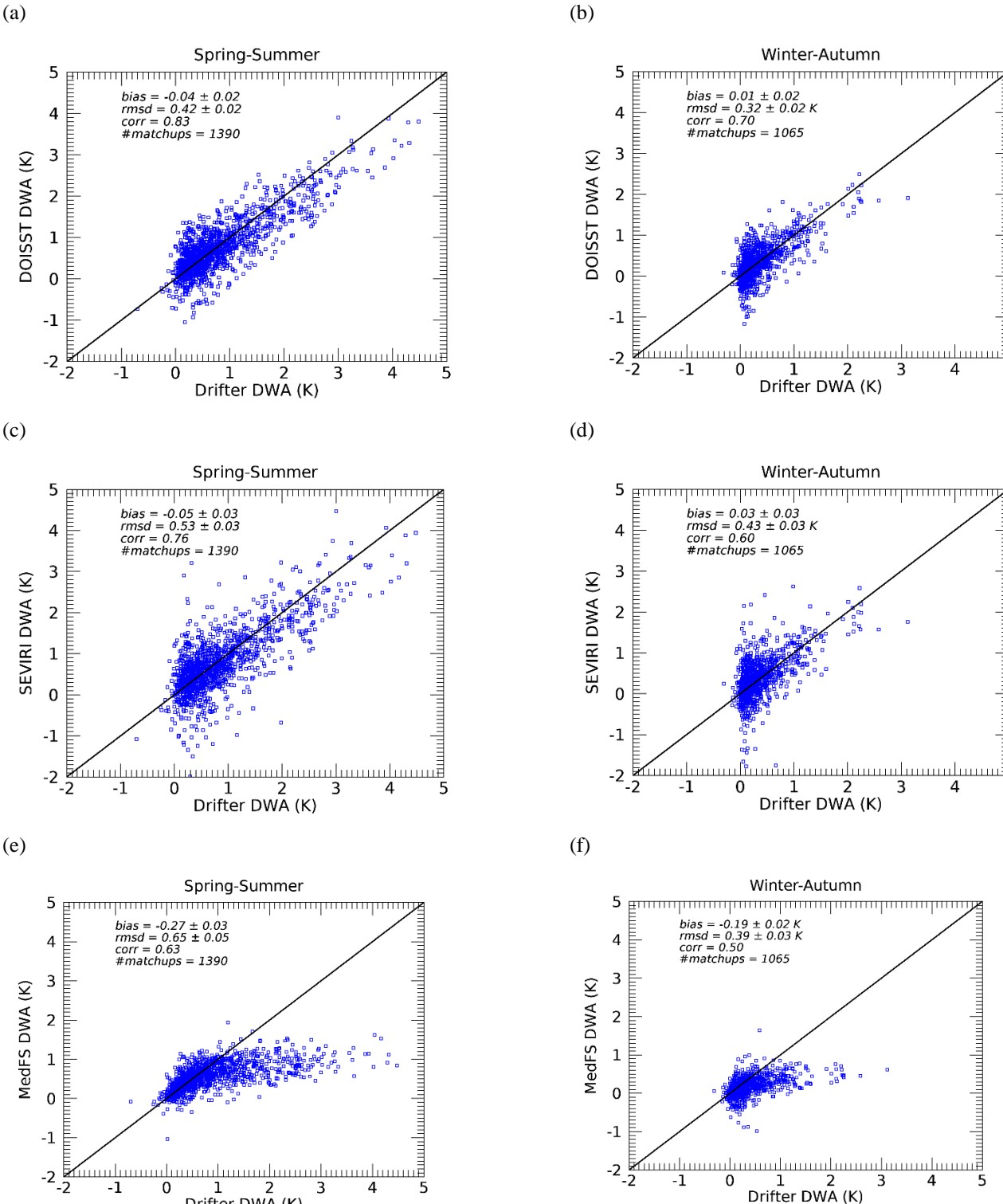

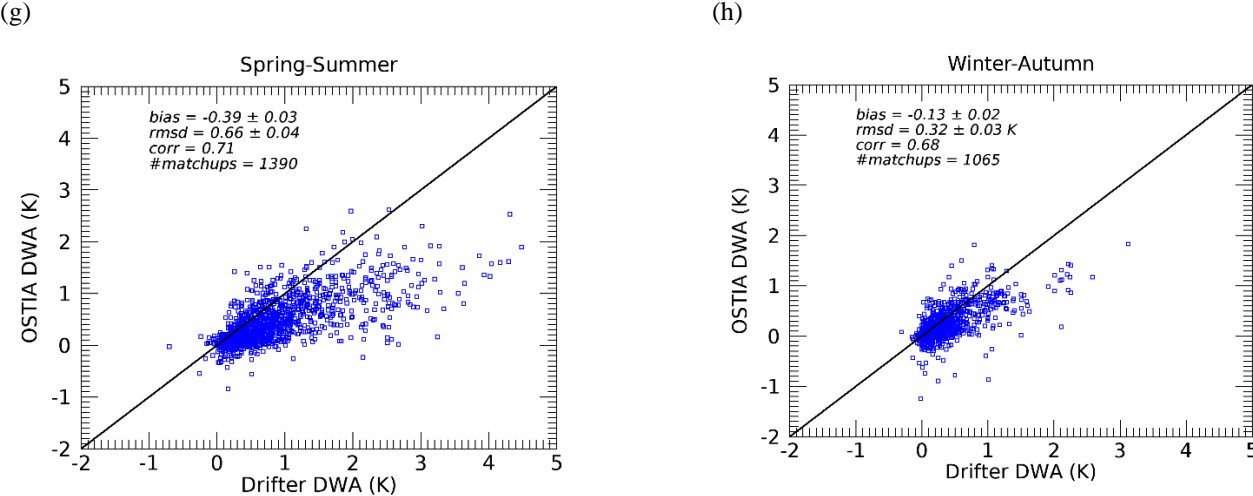

(g)  Spring-Summer  (h)  Winter-Autumn

**Figure 7**. DWA scatter plots for DOISST (a,b), SEVIRI L3C (c,d), MedFS (e,f) and OSTIA diurnal (g,h) vs drifters during Spring (M-A-M) and Summer (J-J-A), and Winter (D-J-F) - Autumn (S-O-N), over the period 2019-2020.

Having demonstrated the reliability of DOISST in the DWA estimate, we analyze its capability to reproduce the typical spatial variability and intensity of DW events in the Mediterranean Sea, a basin characterized by a frequent occurrence of intense DW events (Böhm et al., 1991; Buongiorno Nardelli et al., 2005; Gentemann et al., 2008; Merchant et al., 2008). In our investigation area, the 2019-2020 mean DWA ranges from a minimum of 0.4 K in the Atlantic Ocean box off the Strait of Gibraltar, to a maximum of 1.2 K in several regions of the Mediterranean Sea (Fig. 8a) where individual diurnal warming events exceeding 1 or even more than 2 K are quite frequent. The largest DWA were observed in the Levantine Basin, in the North Adriatic Sea and in correspondence with the Alboran Gyre. Less intense, though still remarkable, mean DWA patches reaching 0.9 K are found around the southern tip of the Italian Peninsula as well as in the coastal Ligurian Sea. In the same areas, it is found that the frequency of DW events larger than 1 K and 2 K can reach up to 55% and 10% of the analyzed time series, respectively (bearing in mind that our time series is given by the total number of days in 2019 and 2020) (Fig. 8b-c). The spatial variability and magnitude of the DWA described by the DOISST product are consistent with past and recent studies on the SST diurnal variability in the Mediterranean Area (Minnet et al. 2019; Marullo et al. 2016; Marullo et al. 2014).

The magnitude of the maximum SST diurnal oscillation is also investigated. The spatial distribution of the maximum DWA observed through 2019-2020 in the Mediterranean Sea (6°W to 36°E and 30°N to 46°N) (Fig. 8d) shows that the largest amplitudes reach and exceed 3 K in 98% of the basin and local DWA patches exceeding 6 K are also ubiquitous, confirming that the Mediterranean is one of the areas with the largest DWs of the global ocean (Minnet et al. 2019, and references therein).

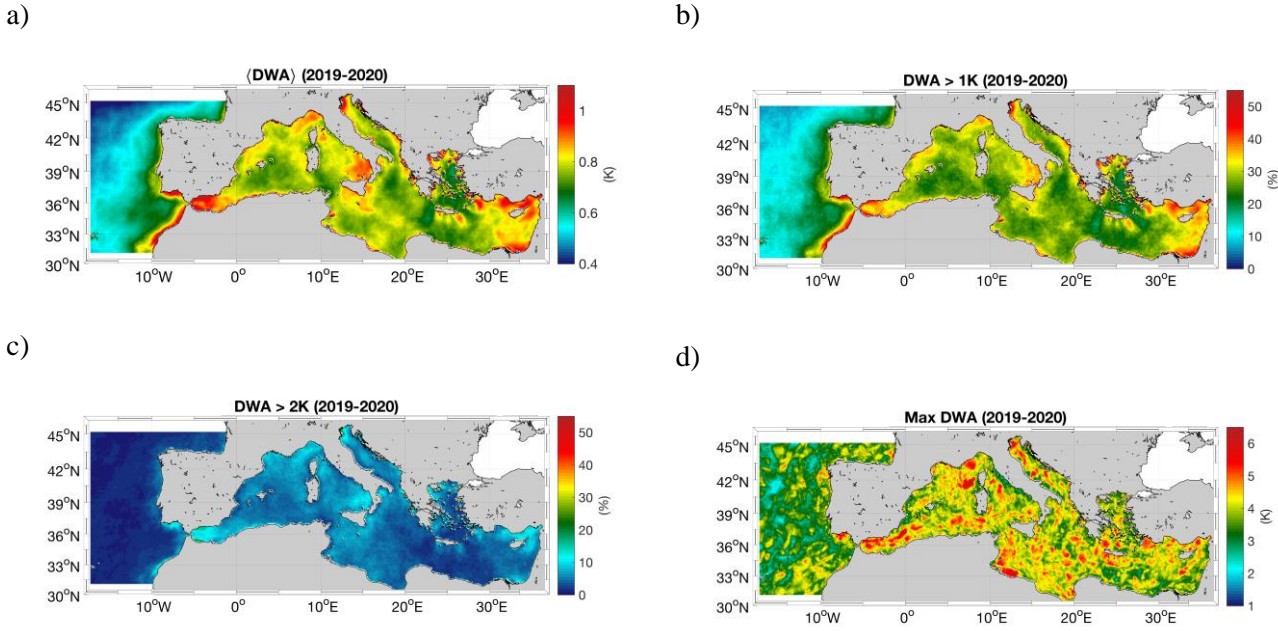


**Figure 8.** a) Mean diurnal warming amplitude (DWA) derived from DOISST; b) Percentage (over the total number of days in
the 2019-2020 period) of DOISST DWA larger than 1 K; c) Percentage of DOISST DWA larger than 2 K; d) Maximum
observed DOISST DWA. All the maps refer to the 2019-2020 period.

When compared to the model, DOISST exhibits mean DWAs with larger intensity than MedFS ones in all the locations of the
study area (Fig. 9). The $\Delta$DWA, defined as DWA $_{DOISST}$ minus DWA$_{MedFS}$, is always larger than 0.2 K and locally reaches
extreme values of ~1 K. The extent of the $\Delta$DWA generally increases in areas where the DOISST mean DWA is larger, such
as in the Alboran Sea, Ligurian Sea, Levantine Basin and Southern Tyrrhenian, suggesting a tendency of the model to
underestimate the largest DW events.

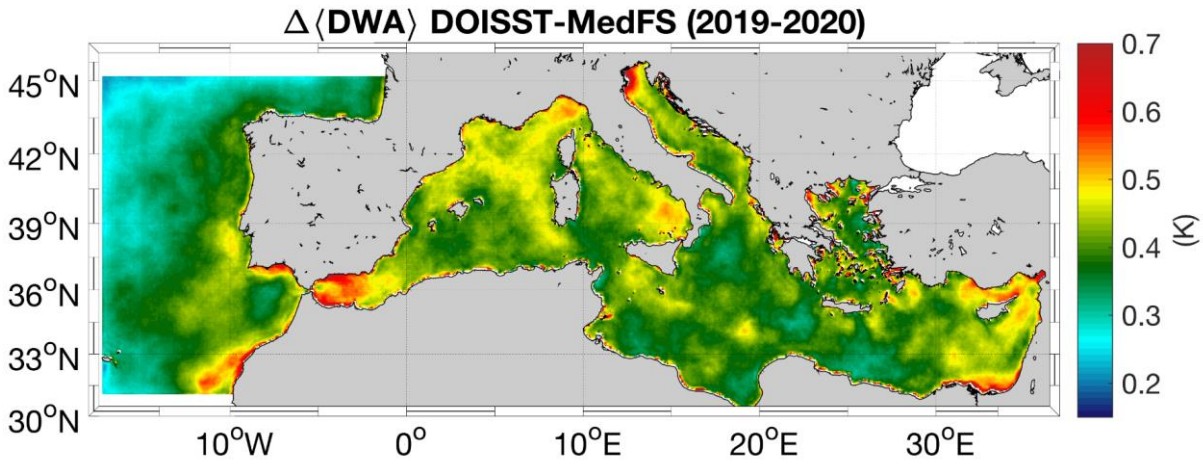

**Figure 9.** Mean amplitude of the SST DW. Differences between the mean DWA seen by DOISST and MedFS.

## 5 Data availability

The Mediterranean diurnal optimal interpolated SST product is distributed as part of the Copernicus Marine Service catalogue, and identified as SST_MED_PHY_SUBSKIN_L4_NRT_010_036 (Copernicus product reference) and cmems_obs-sst_med_phy-sst_nrt_diurnal-oi-0.0625deg_PT1H-m (Copernicus dataset reference) (https://doi.org/10.48670/moi-00170; last access: 15 July 2022; Pisano et al, 2021). Access to the product is granted after free registration as a user of the Copernicus Marine Service at https://resources.marine.copernicus.eu/registration-form (last access: 15 July 2022). Once registered, users can download the product through a number of different tools and services, including the web portal Subsetter, Direct-GetFile (DGF) and FTP. A Product User Manual (PUM) and QUality Information Document (QUID) are also available as part of the Copernicus documentation (https://resources.marine.copernicus.eu/product-detail/SST_MED_PHY_SUBSKIN_L4_NRT_010_036/DOCU MENTATION, last access: 15 July 2022). Eventual updates of the product will be reflected in these documents. The basic characteristics of the DOISST product are summarized in Table 2. The reduced subset used here for validation and review purposes is openly available at https://doi.org/10.5281/zenodo.5807729 (Pisano, 2021).

## 6 Summary and conclusions

A new operational Mediterranean diurnally varying SST product has been released (May 2021) within the Copernicus Marine Service. This dataset provides optimally interpolated (L4) hourly mean maps of sub-skin SST over the Mediterranean Sea at 1/16° horizontal resolution, covering the period from 1st January 2019 to near real time (1 day before real time) (Pisano et al., 2021). The diurnal optimal interpolated SST (DOISST) product is obtained from a blending of hourly satellite (SEVIRI) data and model (MedFS) SSTs via optimal interpolation, where the former are used as the observation source and the latter as background. This method has been firstly proposed by Marullo et al. (2014), validated over one year (2013) in Marullo et al. (2016), and implemented here operationally. The validation of the operational product was also extended over two years (2019-2020) and based on a direct comparison with in situ surface drifting buoys data.

In an ideal case, all data (satellite, model and in situ) would be available at the same depth. Unfortunately, the first MedFS model layer is centered at 1 m depth, while sub-skin SST is, by definition, representative of a depth of ~1 mm. In principle, it could be possible to correct all the data, bringing them all to the same depth before any comparison or merging, by applying some model (see e.g. Zeng et al., 1999). However, any correction algorithm would have added potential uncontrolled error sources (e.g., related to ancillary data and/or to model assumptions) and implied significant additional operational efforts. For these reasons, rather than trying to correct the first-guess bias, we preferred to leave it uncorrected, and focus on optimising the corrections driven by available hourly satellite data.

DOISST proved to be rather accurate when compared to drifter measurements, and correctly reproduced the diurnal variability in the Mediterranean Sea. The accuracy of DOISST results in an overall, almost null, mean bias of ~0.04 K and RMSD of ~0.41 K (Table 3). This product is also more accurate than the input MedFS, which shows a mean bias of ~-0.1 K and RMSD of ~0.47 K. A warm (positive) and cold (negative) bias characterizes the DOISST and MedFS, respectively, also during seasons (Fig. 5). These opposite biases are likely related to the different nature of the SST provided by DOISST, MedFS and drifter data, i.e. sub-skin (~1 mm from the surface), averaged 1 m depth and 20 cm depth, respectively, and then consistent with the physical consequence of a reduction of the temperature with depth due to the vertical heat transfer. The DOISST RSMD generally keeps lower values compared to MedFS, ranging from a minimum of ~0.40 K (vs ~0.42 K for MedFS) to a maximum of ~0.44 K (vs ~0.56 K for MedFS). These results also confirm the robustness of this blending algorithm that, even if based on model analyses used as first-guess, it successfully brings DOISST closer to the in situ measured SST than the MedFS estimates.

Compared to its native version (Marullo et al., 2016), the DOISST product maintains the same RMSD (estimated in 0.42 K) but displays a lower mean bias (estimated as -0.10 K). The reduced bias could be ascribed to the fact that valid SEVIRI SST values are always interpolated in DOISST, while they are left unchanged (not interpolated, see section 3.3) in the original method. Additionally, the DOISST bias is comparable with that estimated for SEVIRI over the Mediterranean Sea (-0.03 K; Marullo et al. 2016), while the DOISST RMSD is rather lower than SEVIRI one (0.47 K; Marullo et al. 2016). The DOISST

bias is also lower than that of the OSTIA diurnal product, which produces gap-free hourly mean fields of skin SST for the
global ocean, and has been found to underestimate the diurnal range of skin SST by 0.1-0.3 °C (While et al., 2017).
The analysis of the SST diurnal cycle as estimated from both DOISST, MedFS and drifter data shows that the diurnal oscillation
in SST is well reconstructed by the DOISST while MedFS tends to underestimate this amplitude mainly during the central
warming hours (Fig. 4), and during spring and summer (Fig. 5b, c). Specifically, DOISST overestimates the mean diurnal
amplitude by ~2.3% compared to that of drifters, while MedFS underestimates it by ~16%. This is particularly evident in the
analysis of diurnal warming (DW) events, where diurnal warming amplitudes (DWAs) as estimated by DOISST, MedFS,
SEVIRI, and OSTIA diurnal data are compared vs drifter-derived DWAs. This analysis shows that amplitudes exceeding 1 K,
as measured by drifters, are well reconstructed by DOISST (Fig. 6a) with a mean bias of ~-0.02 K and RMSD of ~0.38 K. The
comparison with reconstructed SEVIRI DWAs (Fig. 6b) demonstrates that optimal interpolation does not change the SEVIRI
bias, which is practically null for both SEVIRI and DOISST (~-0.02 K), while it reduces the SEVIRI RMSD, from ~0.49 K
(SEVIRI) to ~0.38 K (DOISST). This is also evident in the reduction of the spread of SEVIRI DWAs around the line of perfect
agreement (Fig. 6b). Both MedFS and OSTIA diurnal underestimate DWAs when exceeding 1 K with a mean bias of ~-0.23
K (MedFS, Fig. 6c) and ~-0.28 K (OSTIA, Fig. 6d), and RMSD of ~0.55 K for both products. This underestimation could be
related to several factors, such as that the vertical resolution of MedFS does not resolve the vertical temperature profile within
the warm layer. Yet, the physics and atmospheric forcing and/or the assimilation implemented in MedFS and OSTIA, though
different, are only partially able to resolve diurnal variations larger than 1 K. In any case, we can argue that the tendency of
MedFS to underestimate DWAs, mainly for amplitudes > 1 K, does not strongly impact the performance of DOISST in
reconstructing these amplitudes. This is likely due to two concurrent factors, the high accuracy of SEVIRI SST data and that
the Mediterranean area is particularly advantageous in terms of clear sky conditions.
Finally, the seasonal analysis also reveals that DOISST is not impacted by the different environmental conditions in the
Mediterranean Sea, in particular from the much frequent cloudiness during winter and autumn periods.
Overall, the DOISST product is able to accurately reconstruct the SST diurnal cycle, including diurnal warming events, for the
Mediterranean Sea and can thus represent a valuable dataset to improve the study of those processes that require sub-daily
frequency.

**Financial Support**
This work has been carried out within the Copernicus Marine Environment Monitoring Service - Sea Surface Temperature
Thematic Assembly Centre (SST TAC), contract n° 78-CMEMS-TAC-SST. This contract is funded by Mercator Océan
International as part of its delegation agreement with the European Union, represented by the European Commission, to set-
up and manage the Copernicus Marine Service.

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
