# Peer review of "A New Operational Mediterranean Diurnal Optimally Interpolated"

_Earth System Science Data, 2021_

## Author Comment (AC1)

**RC1**: 'Comment on essd-2021-462', Anonymous Referee #1, 21 Feb 2022 reply

*This study presents a high resolution (1/16-deg and 1-hr) DOISST using SEVIRI satellite as source and a model SST as first-guess, which will no doubt have many applications. Their analysis demonstrated that DOISST well represent the diurnal cycle with a low mean bias and RMSD of 0.4ºC. The seasonal features of SST diurnal cycle in MED area are described and compared with independent buoy SST and model simulation. My major concern is how the model SST at 1 m depth is used as the first-guess field. It is not clearly stated what depth the DOISST represent, and how the DOISST is validated with buoy SSTs at 0.2 m depth. The ideal case is that all three components are compared/generated at the same depth level. I recommend accept the manuscript after a major revision.*

First, we thank the reviewer for the stimulating comments and questions. Overall, the reviewer helped us understand that several key aspects lacked clearness and detail and we do agree that they need to be introduced and described with more detail. This could partly be originated from having taken for granted some of these aspects, as e.g. the different types of SST definitions and some commonly adopted validation procedures. In any case, following the reviewer's comments, we worked to improve the level of detail.

We will start answering the more general comments/questions of the reviewer, then we will provide answers to the more specific comments.

*"My major concern is how the model SST at 1 m depth is used as the first-guess field".*

The first model layer is unfortunately centered at 1 m depth, so we used the SST at this level to produce an anomaly field that represents the difference between the observed hourly satellite and our first-guess. This anomaly is the variable that we interpolate over data voids using our Objective Interpolation (OI) scheme. This first-guess choice is a better alternative to the use of climatologies or previous analysis data, as operated by other schemes in producing daily SST L4 maps (see Marullo et al., 2014), since it gives the best estimate of hourly SSTs in the absence of any observation or in situ measurement. This choice simplifies the computation of the space-time covariance function that is used to weight the input observations within the OI algorithm. In fact, figure 3 of Marullo et al. (2014) shows the behavior of the correlation function versus time when either the hourly model or the SST daily climatology are subtracted. It is evident that, in the second case a strong daily component is present, while in the first case, the daily component is significantly reduced, allowing to state that the data are "nearly" free from the diurnal cycle. Anomalies observed at different times can thus be combined to better describe the diurnal warming patterns. Indeed, as we aim to retrieve in the most accurate way the spatial pattern of the surface warm anomalies (which evolves over time), it is desirable that observations that are closer in time are weighted more by the OI algorithm (through covariance) than distant ones.

Of course, we do agree that it would have been better to use a sub-skin model SST as first-guess but, at the present time, such a product is not available. In principle, it could be possible to correct all the data, bringing them all to the same depth before any comparison or merging, by applying some model (see e.g. *Zeng et al., 1999). However, any correction algorithm would have added potential uncontrolled error sources (e.g., related to ancillary data and/or to model assumptions) and implied significant additional operational efforts. For these reasons,

rather than trying to correct the first-guess bias, we preferred to leave it uncorrected, and focus on optimising the corrections driven by available hourly satellite data.

[Figure]

**Fig. 3.** Covariance structure function of the Mediterranean Sea estimated from summer 2011 SEVIRI data. (a) Temporal covariance: in red the covariance is computed using hourly SST anomaly field (hourly SEVIRI–hourly MFS); in black the covariance is obtained using hourly SST field after removal of mean daily SST (hourly SEVIRI–mean daily MFS). Vertical bars represent ± 1 standard deviation. (b) Spatial covariance function at Δt = 0 for the SST anomaly field. Dotted blue curves in (a) and (b) represent Eqs. (1) and (2) respectively.

These concepts have been included in the revised manuscript (see e.g. Abstract and Conclusions). In particular, we added Section 3.2, which better introduces the optimal interpolation method and the choice of a model output as first-guess.

*Zeng, X., Zhao, M., Dickinson, R. E., & He, Y. (1999). A multi-year hourly sea surface skin temperature dataset derived from the TOGA TAO bulk temperature and wind speed over the tropical Pacific. Journal of Geophysical Research, 104, 1525–1536.

*"It is not clearly stated what depth the DOISST represent, and how the DOISST is validated with buoy SSTs at 0.2 m depth".*

Actually, the depth information is intrinsically contained in the definition of the SST provided by DOISST, namely the sub-skin SST. "This product provides hourly mean maps (Level-4) of sub-skin SST" (as stated now at line 12), and "sub-skin SST is the temperature at the base of the conductive laminar sub-layer of the ocean surface, as defined by the Group of High Resolution SST (GHRSST) (line 91; see also figure 1, and Minnett et al., 2019). In practice, this is the temperature at ~1 mm depth" (line 93). Here, the lack of clarity could be due to the fact that we did not highlight that SEVIRI provides sub-skin SST (as also reported in the SEVIRI product user manual (PUM), see https://osi-saf.eumetsat.int/lml/doc/osisaf_cdop3_ss1_pum_msg_sst_data_record.pdf), and, being SEVIRI the predominant input into the optimal interpolation scheme, for consistency our DOISST provides sub-skin SST. In section 3.1, we added a paragraph that clarifies the depth represented by DOISST. We also added in Table 2 the depth information of the DOISST product. Finally, a clear reference to this depth was added in the Abstract too.

Of course, the differences in observations' representativeness make it difficult to compare in situ with satellite data, since there are no in situ instruments able to routinely measure skin/sub-skin sea surface temperatures. The commonly adopted validation procedure is in fact to use surface drifting buoys due to both their high accuracy and closeness to the sea surface, namely at ~20 cm. Of course, also these observations include a representativeness error when compared

to sub-skin SST estimates. This concept has been introduced in the revised text (see lines 171-176).

[Figure]

Fig. 2. Near-surface oceanic temperature gradients. From Minnett and Kaiser-Weiss (2012).

Figure 1. The different types of SST based on the GHRSST definitions (GHRSST - Products).

*"The ideal case is that all three components are compared/generated at the same depth level."*

We definitely agree with the referee; the ideal case would be if all data were generated and compared at the same depth. Unfortunately, SST data are instead acquired at different depths and with a variety of instruments of different efficiency and precision. As also suggested by the reviewer, it could be possible to correct all the data, bringing them all to the same depth before any comparison or merging, by applying some model. However, as already stated in a previous answer, any correction algorithm would introduce additional potential error sources rather than eliminate a (small) bias due to depth differences. That's why we preferred to leave the near surface temperature first-guess data uncorrected, and focus on optimising the corrections driven by available hourly satellite data.

This concept is introduced in the new Section 3.2 and Conclusions (lines 536-542) of the revised manuscript.

*"L14, it is not clear why the model analysis is used as the first-guess".*

The model takes into account the effect of air-sea interactions by imposing external forcings that drive momentum and heat exchanges at the upper boundary. As such, it is able to reproduce at least part of the diurnal warming effects, that are driven by the forcing diagnosed from atmospheric model analyses. Using the model output as a first-guess thus allows to treat the hourly SEVIRI data as corrections to the hourly model data. These anomalies are generally

small and mostly describe corrections to the spatial patterns, while displaying a reduced diurnal cycle. Anomaly data from different times of the day can thus be more "safely" used to build the interpolated field at each reference time (with different weights). Unfortunately, the first model layer is at 1 m depth, which means that it will generally underestimate the diurnal cycle anyway. While 1D models could in principle be used to better reproduce sub-skin SST from model data, the approach presented here is focusing on providing estimates that are much closer to original satellite data, avoiding the complications of setting up an additional preprocessing step just to improve the first-guess.

This concept is entirely reported in the new Section 3.2.

*L15-16, it is not clear what is "any diurnal cycle". "differences between satellite and model SST are free, or nearly free". If this is the case, then why do we need DOISST analysis?*

The anomaly we are looking at is computed as the difference between the satellite derived hourly SST field and the model hourly analysis (see Figure 2 from Marullo et al., 2014). This anomaly is the variable to be interpolated. If the model outputs were representing the same layer sensed by SEVIRI and model physics at the air-sea interface was accurately represented, this anomaly would be free from the diurnal cycle. See also previous answer to the question "*My major concern is how the model SST at 1 m depth is used as the first-guess field*".

However, we do agree with the reviewer, that sentence is rather unclear. We re-written this sentence as follows (see lines 16-18): "The choice of using a model output as first-guess represents an innovative alternative to the commonly adopted climatologies or previous analyses, providing physically consistent estimates of hourly SSTs in the absence of any observation or in situ measurement".

[Figure]

**Fig. 2.** Example of model first-guess and original SST SEVIRI data over a point in the Mediterranean Sea. (a) Model (black curve) and SEVIRI (red curve) SST time series. (b) Difference between the SEVIRI and model SST.

*"L35, Does this mean, the SST analysis will be absent when it is rain?"*

Satellite-based SST images are frequently, and usually, affected by several data voids since infrared and microwave sensors cannot "see" under cloudy and rainy conditions. Therefore, many applications require SST data to be processed up to what is generally called Level 4 (L4),

namely gap-free fields. The DOISST is an example of such a product, obtained through statistical optimal interpolation technique.

*"L71, "slightly less than that" => approximately"*

Corrected.

*"L155, SST at 1 m level. What depth does the satellite SST in section 2.2 represent? How is the model SST at deeper level used as a first-guess of the satellite SSTs near the skin level? "*

The CMEMS Mediterranean daily product provides (nighttime) gap-free maps of foundation SST. By definition, foundation SST is the water temperature at a depth such that the daily variability induced by the solar irradiance is negligible. For this reason, a fixed depth can not be assigned a priori, as it changes continuously. A reference value is given at about 10 m (see GHRSST definitions) since, on average, the diurnal warming is not seen anymore at this depth.

The answer to the last question was given in the answer to the previous question *"The ideal case is that all three components are compared/generated at the same depth level"*.

*"L158, how is the buoy SST at 20 cm level used to validate DOISST"*

Here, the answer is the same as that given to the previous question: *"how the DOISST is validated with buoy SSTs at 0.2 m depth"*

*"L166, delete an extra space".*

Corrected.

*L169, "between" => among?*

Corrected

*"Table 1, sub-skin SST, What is the level of sub-skin? please add a depth level".*

Added in Table 2 of the revised manuscript.

*"L188-190, the statement is not clear, and need to be clarify, particularly "allowing to interpolate SST anomalies using satellite data". Is the "anomaly" referenced to an hourly climatology, how is the climatology is defined?"*

The new section 3.2 should hopefully clarify this point. However, as answered to the first question ("*My major concern is how the model SST at 1 m depth is used as the first-guess field*"), the choice of a model output as first-guess allows to use a covariance function which is practically monotonic decreasing, which, in turn, allows to interpolate SST anomalies at different times of the day (specifically, ± 24 hours around the interpolation time). This could not be achieved by using a climatology, since the covariance function would present local maxima and minima, oscillating during time. Then, no climatology is used, neither introduced in the text.

*"L201-202, what is the difference between L3C SST and L3C sub-skin SST? How are the SSTs at different levels blended in DOISST"?*

There is no difference. Indeed, L3C indicates the processing level (namely, single-sensor collated file) while sub-skin indicates the type of SST. In other words, OSI-SAF routinely processes SEVIRI measurements providing L3C maps of sub-skin SST, which are downloaded by our DOISST processing system and used to produce the DOISST product. This is explained in section 2.1. The blending of hourly L3C data is obtained through optimal interpolation, and detailed in section 3.2.

*"L212, f(r,dt)=f(r)*(dt) may not be appropriate. Delete "f(r)*(dt)="?"*

Deleted.

*"L224, "no first-guess data are used", how is this possible as described in L227-234?"*

We agree with the reviewer, this sentence is actually not clear. Overall, the first-guess is always subtracted to observations to create anomalies. However, as stated at line xx, after the interpolation, the first-guess is added back to the optimally interpolated anomalies to get the actual SST value. Then, if at least one observation is present within the spatial and temporal bounding box of the interpolation pixel, first-guess pixel value is corrected. We have rephrased the unclear sentence as follows: "This error ranges between 0-100%, meaning that the error is almost zero when an optimal number of observations is present within the space-time influential radius, while only first-guess data are used (i.e. no observations are found within the search radius) when the error is 100%." (see lines 312-314).

*"L240, how is "co-located" defined, interpolated to the in situ location and time or rounded to a certain spatial and time resolution?"*

As stated at line 337: "…the validation is carried out on hourly basis, building a matchup database by collecting the closest (in space) SST grid point to the in situ measurement within a symmetric temporal window of 30 minutes with respect to the beginning of each hour". We substituted "validation" with "co-location" to make this sentence clearer.

*"L272-273, How is the uncertainty of RMSD (±number) calculated?"*

This was actually stated (now at lines at lines 348-349): "Validation statistics are quantified in terms of mean bias and Root-Mean-Square Difference (RMSD) from matchup temperature differences (namely, SST minus drifter). Each statistical parameter is associated with a 95% confidence interval computed through a bootstrap procedure (Efron 1994)". For clearness, we added in each caption this reference.

*"L295, is it possible the biases result from the first-guess of the model SST at different level?"*

It is possible, but very marginally. Indeed, DOISST is the result of a blending of SEVIRI sub-skin SSTs, representative of a depth of 1 mm, and modeled SSTs at 1 m. Then, the DOISST effective depth does, in principle, vary between 1 mm up to 1 m, depending on how many satellite observations enter the interpolation. As diurnal warming is significantly reduced under cloudy conditions, however, the difference between the SST at 1 m and the sub-skin SST will be much smaller when SEVIRI observations are not present. For this reason, we can define the DOISST product as representative of sub-skin values. We added this concept in Section 3.1 (lines 237-242).

*"L337, define DWA earlier"*

Thanks for notifying, this has been corrected.

*"L347-347, will the underestimation of DWA in model affect the performance of DOISST since it is used as the first-guess?"*

Overall, the DOISST improves the description of the diurnal cycle, including DWA estimates, with respect to a purely model-derived estimate. As shown in Section 4.2.2, while the model clearly underestimates diurnal amplitudes larger than 1 K, the DOISST is able to correctly reconstruct these amplitudes. This is likely due to two concurrent factors, the high accuracy of SEVIRI SST data and the fact that the Mediterranean area is particularly favorable in terms of clear sky conditions. Then, we could argue that the underestimation of modelled DWAs does not strongly impact the performance of DOISST. This concept was added in the revised manuscript (see line 574-577 in Conclusions).

*"L385-386, it may be better to explain the reasons".*

We thank the Reviewer for this comment. Firstly, in our manuscript we documented that, in general, the model outputs tend to underestimate the SST diurnal warming (DW) with respect to the DOISST. Investigating the spatial variability of such underestimations, we expected the model to produce weaker diurnal cycles in the areas where this signal is known to be intense (see e.g. the southern Tyrrhenian and the east Mediterranean, north of Cyprus), in very good agreement with previous results described in Marullo et al. 2016. Roughly speaking, in open ocean contexts, the diurnal cycle is modulated by the relative role of insolation and wind-induced mixing. From figure 8, one can see that the DW tends to be larger in areas sheltered by the strongest wind systems or in correspondence of freshwater/lower salinity discharge areas (Zecchetto & De Biasio 2007, Minnet et al. 2019, Field 2007). In our study, this is visible in the southern Tyrrhenian, the east Mediterranean area north of Cyprus, in correspondence of the Po/Nile rivers deltas and also in the Alboran gyre (i.e. in correspondence of the low salinity Atlantic Water inflow). However, to document the mechanisms behind the DW is out of the scope of the present study, where we mostly present the DOISST production and quality assessment. We thus suggest the readers refer to previously published papers on this topic (duly documented in our manuscript) for further insights on the DW mechanisms (see lines 490-492 of the revised manuscript).

*"L422, these depth information should be presented much earlier"*

This is just a reminder. The definition of sub-skin and its depth is actually given much earlier, at line 90 and now in the Abstract too.

*"L429, In MED area or over the global oceans?"*

Those values have been found for validation in the global ocean. However, we also performed an intercomparison between OSTIA diurnal and DOISST in the Mediterranean Area (see revised paper, figures 6-7). Such analysis shows that DOISST yields a more accurate description of the diurnal variability than OSTIA in the Mediterranean Area.

---

## Author Comment (AC2)

**RC2**: 'Comment on essd-2021-462', Anonymous Referee #2, 22 Feb 2022 reply

*The paper describes a new sea-surface temperature product merging SEVIRI data and results from a data-assimilative model. The important novelty is that this product resolves the diurnal cycle and provides full fields without gaps (level 4). The authors also include a quite detailed comparison with in situ observations. While this paper describes the results for the years 2019 and 2020, this data set is continuously updated and the results for the year 2021 are also available.*

*The main questions that I asked myself while reading the manuscript are:*

*A. As there are different depths for the different types of SST (skin-temperature, bulk temperature, foundation temperature) which depth level is the DOISST targeting by this product? I understood that the model and SEVIRI data have different reference depths. Should there not be first a conversion/adjustment, so that the temperature is comparable? Maybe interpreting some of the conclusions within this context would be useful.*

DOISST is indeed the result of a blending of SEVIRI sub-skin SSTs, representative of a depth of 1 mm, and modeled SSTs at 1 m. Then, the DOISST effective depth does, in principle, vary between 1 mm up to 1 m, depending on how many satellite observations enter the interpolation. As diurnal warming is significantly reduced under cloudy conditions, however, the difference between the SST at 1 m and the sub-skin SST will be much smaller when SEVIRI observations are not present. For this reason, we can define the DOISST product as representative of sub-skin values. We added this concept in Section 3.1 of the revised manuscript (lines 237-242). We also added in Table 2 the depth level of the DOISST product.

Concerning the second question, we definitely agree with the referee. The ideal case would be if all data were generated and compared at the same depth. Unfortunately, the first model layer is centered at 1 m depth, while sub-skin SST is, by definition, representative of a depth of 1 mm. In principle, it could be possible to correct all the data, bringing them all to the same depth before any comparison or merging, by applying some model (see e.g. Zeng et al., 1999). However, any correction algorithm would have added potential uncontrolled error sources (e.g., related to ancillary data and/or to model assumptions) and implied significant additional operational efforts. For these reasons, rather than trying to correct the first-guess bias, we preferred to leave it uncorrected, and focus on optimising the corrections driven by available hourly satellite data. This concept has been added in Summary and Conclusions (lines 536-542).

Additionally, a new section (3.2 in the revised manuscript) should clarify this point. As written from line 258: "The choice of using a model output as first-guess represents the best alternative to the use of climatologies or previous analyses, as usually done by other schemes to produce daily SST L4 maps, since the model provides physically consistent estimates of hourly SSTs in the absence of any observation or in situ measurement (Marullo et al., 2014). In fact, the model takes into account the effect of air-sea interactions by imposing external forcings that drive momentum and heat exchanges at the upper boundary. As such, it is able to reproduce at least part of the diurnal warming effects, that are driven by the forcing diagnosed from atmospheric model analyses. Using the model output as a first-guess means we are treating the hourly satellite data as corrections to the hourly model data. These anomalies are generally small and mostly drive corrections to the spatial patterns, while displaying a reduced diurnal cycle. Anomaly data from different times of the day can thus be more "safely" used to build

the interpolated field at each reference time (with different weights). Unfortunately, the first model layer is at 1 m depth, which means that it will generally underestimate the diurnal cycle anyway. While 1D models could in principle be used to better reproduce sub-skin SST from model data, the approach presented here is focusing on providing estimates that are as close as possible to the original satellite data, avoiding the complications of setting up an additional preprocessing step just to improve the first-guess."

*Zeng, X., Zhao, M., Dickinson, R. E., & He, Y. (1999). A multi-year hourly sea surface skin temperature dataset derived from the TOGA TAO bulk temperature and wind speed over the tropical Pacific. Journal of Geophysical Research, 104, 1525–1536.

*B. Comparison: I would have expected a comparison to show that DOISST is better (compared to in situ observations) than other observational products. However, the author compared the new product to a model solution. Are there other data L4 products available (resolving the diurnal cycle) based on SST data from geostationary satellites? In any case, the authors also compare the accuracy of their product (relative to drifters) to the accuracy of the SEVIRI data (at exactly the same location) which already shows some quite favorable results.*

We do agree with the reviewer and thus added the global operational diurnal L4 SST OSTIA product in our intercomparison exercise. In particular, as OSTIA ingests the in situ data we used as reference for the validation (which would not be independent for OSTIA), we included OSTIA diurnal in section 4.2.2, which is dedicated to the reconstruction of diurnal warming amplitudes (DWAs) from different sources (DOISST, Model, SEVIRI and In situ data). The first part of the validation (section 4.2.1) is indeed mainly thought to assess the accuracy of the DOISST product against an independent in situ data source, and the inclusion of modelled SST data is thought to evaluate the DOISST performance with respect to the model, which is used as first-guess.

*I recommend publications after minor revisions.*

*Minor comments:*

*1. line 106: assessment of the MED DOISST product covers two complete years (2019-2020). Please clarify earlier in the manuscript the time coverage of the data product and the time coverage of the assessment.*

Clarified (see lines 114-117).

*2. degree K (line 19, abstract) or degree C (line 39, introduction). Can you please use the same units?*

Corrected.

*3. an overview table with all products would be useful, including resolution (time and space) and coverage (time and space) and reference depth (e.g. skin, subskin, foundation temperature,...), even if the study uses a subset of the input data set. This table could also include the new dataset.*

An overview Table has been added (i.e., Table 1 of the revised manuscript).

*4. typesetting of the equation should be improved and follow the style of other Copernicus papers.*

This has been corrected.

*5. page 9: "All these parameters have been deduced from a statistical analysis of the satellite SST data" Please give more information about how you choose the particular parameters (a, c, d, decorrelation spatial length R, decorrelation time length T). In particular, what objective criterion was used to decide that these parameters are appropriate?*

This has been clarified (see lines 302-303): "All these parameters have been derived in Marullo et al. (2014), deduced from a nonlinear least square fit between the estimated temporal and spatial correlations."

*6. page 10, line 250: "At each step of decreasing n, data that falls out of the interval I = [mean(delta) - n sigma, mean(delta) + n sigma] are flagged. The process starts for n=10 and stops at n=3." If the data is outside of the interval for n=3, why would one also check for n=10? But I guess that delta (the difference, and the mean and standard deviation) also depends on n by selecting a different subset for different n. I think that this should be clarified in the proposal.*

This sentence is actually unclear. It has been re-written. See lines 339-346 of the revised manuscript.

*7. line 295: "The two diurnal cycles are practically coincident between 17:00 and 06:00, while they are biased by ~0.1 K between sunrise and 16:00, coherently with the DOISST bias oscillation (Fig. 3). This bias could be related to skin SST getting warmer faster than 20 cm temperature"*

*I suggest you replace "20 cm temperature" by "temperature at 20 cm depth".*

Replaced.

*I am not sure if "coincident" is the right word. What about saying that the bias is close to zero (DOISST and drifter temperature) as you do not show the diurnal cycles of DOISST and drifter temperature individually.*

Figure 4 shows the mean diurnal SST cycle as reconstructed by DOISST, model and drifters, while the bias is shown in Figure 3. However, ae have substituted coincident with unbiased.

---

## Author Comment (AC3)

**RC3**: 'Comment on essd-2021-462', Anonymous Referee #3, 16 Mar 2022 reply

*The authors present a new Mediterranean Sea regional SST product that reproduces the diurnal cycle. For this, the authors merge the SST from the CMEMS Mediterranean Sea Physical Analysis and Forecasting product with the SST measurements from SEVERI remote sensor, and they apply a methodology that is presented in Marullo et al. (2014). For assessing the actual capability of the resulting SST product to properly capture the skin SST variations, the authors use a set of drifting buoy SST measurements that are typically acquired at 20 cm depth. This is a clear limitation of the assessment of, not only this product, but all the satellite products that aim at reproducing the skin dynamics, there is not in situ data to compare with. In the absence of in situ skin SST measurements, the quality assessment that the authors present here is clear, and they provide evidence that the product is properly capturing the diurnal cycle, or at least that it is capturing it better than the model. So, I think the manuscript deserves its publication in the Earth Science System Data journal.*

*I have some minor comments /questions to the authors.*

*Line 15-16: "The differences between satellite and model SST are free, or nearly free, of any diurnal cycle"->I don't understand this I though model does not reproduce the diurnal cycle while the satellite does*

Both the model and the satellite reproduce a diurnal cycle, but the two cycles are not identical. If they were, their difference (our SST anomaly) would not contain any diurnal signal. In the real case a small difference (mainly in terms of amplitude) still exists. This implies that the SST anomaly contains a small diurnal component. In this sense we can say "Free or nearly free", consistently with fig. 2 of Marullo et al. (2014).

However, we do agree that that sentence is rather unclear. It has been removed and re-written as follows: "The choice of using a model output as first-guess represents an innovative alternative to the commonly adopted climatologies or previous analyses, providing physically consistent estimates of hourly SSTs in the absence of any observation or in situ measurement". We also added Section 3.2 in the revised manuscript, which better introduces the optimal interpolation method and the choice of a model output as first-guess.

[Figure]

Fig. 2. Example of model first-guess and original SST SEVIRI data over a point in the Mediterranean Sea. (a) Model (black curve) and SEVIRI (red curve) SST time series. (b) Difference between the SEVIRI and model SST.

*Line 17: I'm wondering whether these drifting buoys are assimilated in the model or not.*

From the model documentation, available via the Copernicus Marine Service website (https://resources.marine.copernicus.eu/product-detail/MEDSEA_ANALYSISFORECAST_PHY_006_013/DOCUMENTATION), one can see that the assimilated variables are:

- In-situ vertical profiles of Temperature and Salinity from ARGO and XBT;

- Sea Level Anomalies (SLA) from available satellites Jason 2 & 3, Saral-Altika, Cryosat; Sentinel-3A/3B.

- Objective Analyses-Sea Surface Temperature (SST) fields, used to correct surface heat fluxes.

Therefore, drifting buoys can be considered as a fully independent validation benchmark.

*Line 93: It would be interesting for the reader a comparison between the performance of this skin SST OSTIA and MED DOISST.*

We do agree with the reviewer and thus added the global operational diurnal L4 SST OSTIA product in our intercomparison exercise. In particular, as OSTIA ingests the in situ data we used as reference for the validation (which would not be independent for OSTIA), we included OSTIA diurnal in section 4.2.2, which is dedicated to the reconstruction of diurnal warming amplitudes (DWAs) from different sources (DOISST, Model, SEVIRI and In situ data). The first part of the validation (section 4.2.1) is indeed mainly thought to assess the accuracy of the DOISST product against an independent in situ data source, and the inclusion of modelled SST data is thought to evaluate the DOISST performance with respect to the model, which is used as first-guess.

*Line 106: Do the authors plan to extend the temporal series backwards?*

No such a plan at the moment. Future evolutions will mostly depend on users' feedback.

*Lines 128-130: I don't understand this. Why are the differences between SEVERI SST and drifters larger during nighttime than in daytime? I would expect larger differences during daytime because drifter measurements are acquired at 20cms depth and SEVERI measurements are provided in the first mm. Are these differences reflecting in first order the radiometric errors of SEVERI?*

First, these validation results were produced by OSI-SAF, the data provider. From our side, we do not actually see large differences between nighttime and daytime statistics. Indeed, the bias is practically identical during nighttime and daytime, resulting in -0.1K and -0.09K respectively. This means that the bias remains unchanged during the 24 hours, thus revealing a good stability of the SST retrieval. Similar considerations are for the RMSD with the only difference being that the error during daytime (0.56K) is slightly higher than that at nighttime (0.53K).

*Line 166: Delete " " before "."*

Deleted.

*Lines 188-191: I don't understand this paragraph: 1) Why are you using differences between satellite and model instead of satellite measurements directly? I don't understand the point of the reduction of one order of magnitude of the difference. 2) Do you mean that for generating hourly products you are considering all observations around the model in +/- 24 hours? Have you assessed the impact on the final product of considering different (reduced) temporal windows?*

1) The optimal interpolation (or statistical interpolation) method determines the optimal solution to the interpolation of a spatially and temporally variable field with data voids, where "optimal" is intended in a least square sense (see e.g. Bretherton et al., 1976). Optimal interpolation requires two datasets, observations (as satellite-based SST measurements) and first-guess estimates (as model output SSTs). The optimally interpolated variable, or analysis (Fa), is obtained as:

$$F_a(x,t) = F_b(x,t) + \sum_{i,j=1}^{n} W_{i,j}(F_{obs,i}(x,t) - F_b(x,t))$$

In practice, the analysis $F_a$(x,t) at a particular location $(x,t)$ (in space and time) is obtained as a correction to a background field $(F_b(x,t))$, estimated as a linear combination of the observation anomalies $(F_{obs}-F_b)$, where the coefficients $W_{i,j}$ are obtained minimizing the analysis error variance.

The reduction of one order of magnitude means that the difference between satellite and model SST (our SST anomaly) contains only a small amount of diurnal signal.
These concepts are now included in the Section 3.2 of the revised manuscript.

2) Yes, each hourly product is generated by using 24 SEVIRI L3C SST maps, following the approach proposed by Marullo et al. (2014). This choice was actually the result of several trials, and this has been clarified in the revised text (line 307).

*Line 203: I would specify here also the model spatial grid.*

This has been corrected

*Line 204: I would specify at which grid the regridded is performed.*

The remap is performed over a 0.0625° regular grid. We added in the revised text the reference to Table 2.

*Line 256: Estimates of the correlation with in situ may also provide useful information.*

We added estimates for correlation coefficient (see Table 3 in the revised manuscript).

*Line 258: Have you assessed SEVERI SST? It would be interesting for the reader the comparison between SEVERI and MED DOISST performances (not only in the DWA).*

This is a good comment. We decided not to include, at least in this subsection, a comparison with SEVIRI for different reasons. First, the inclusion of modeled SST data in the first part of our validation (section 4.2.1) is mainly thought to evaluate the DOISST performance with respect to the model, which is used as first-guess. This is the same reason for which we included OSTIA diurnal in the second part of the validation (Section 4.2.2). Then, the inclusion of SEVIRI data would have reduced the number of matchups since these data present gaps (data voids). Additionally, the validation for SEVIRI data has already been performed in two previous papers, namely in Marullo et al. 2014, 2016, and we often refer to these papers. However, we added in section 2.1 (lines 142-144) the main results obtained from Marullo et al. (2016) that quantify the bias and RMSD for SEVIRI SSTs over the Mediterranean Sea, and compared them to those obtained for DOISST in the Summary and Conclusions (lines 554-556).

*Line 262: I would say pointwise difference.*

Corrected.

*Fig. 2: Perhaps it would be interesting to separate the map into daytime and night time.*

Interesting suggestion. We separated the computation between day and night but the two corresponding maps are practically identical, indistinguishable. This might not be so surprising since the bias between DOISST and in situ temperatures is always very low, both during nighttime and daytime (see Table 4 of the revised manuscript). For this reason, we think it is preferable to keep just a single map.

*Line 265: "tendency"->"predominance"*

Corrected.

*Fig 6. Is interesting that although the dispersion of DOISST DWA around Drifter DWA hasbeen significantly reduced with respect to the one of SEVIRI DWA, the maxima DWA events seem to be better captured with SEVERI than with DOISST (that they seem to be a bit underestimated).*

Yes, this is quite reasonable since SEVIRI provides a direct measurement of sub-skin SST while DOISST performs a blending of these temperatures with modelled data.

---

## Editor Decision (ED1)

**A New Operational Mediterranean Diurnal Optimally Interpolated SST Product within the Copernicus Marine Environment 2 Monitoring Service**

Andrea Pisano, Daniele Ciani, Salvatore Marullo, Rosalia Santoleri, Bruno Buongiorno Nardelli

A new operational MEDiterranean Diurnal Optimally Interpolated Sea Surface Temperature (MED DOISST) product has been developed within the Copernicus Marine Environment Monitoring Service (CMEMS). MED DOISST is characterized by hourly mean maps (Level-4) of sub-skin SST at 1/16° horizontal resolution over the Mediterranean Sea from January 2019 to present. The sub-skin SST is the sea temperature at ~1 mm depth, which is subject to a large diurnal cycle.

The product is built by blending hourly SST data from SEVIRI and model analyses from the CMEMS MED-MFC system (Clementi et al. 2021) through optimal interpolation. The use of model SST as first-guess substitutes the adoption of climatologies or **previous analyses**, providing improved physically consistent estimates of hourly SSTs.

The manuscript after a first revision has been improved, answering the reviewers comments and it is accepted for further publication. However there are some issues, which I summarize hereafter, that I kindly ask you to take into consideration.

Line 15: I would specify which model dataset (as for SEVIRI) defining a name to keep over the manuscript --> some suggestions: MED-MFC (Clementi et al. 2021) or MedFS or Med-currents.

**Line 18**: "...*in the absence of any observation or in situ measurement*" This phrase is misleading since there are surface drifting buoys data that you use for validation. Why didn't you blend SEVIRI with in situ data? Could you please clarify?

**Line 29**: This statement should be clarified, how would it improve the model predictability? Would you assimilate or use this product to correct the heat fluxes in other general ocean circulation models or do you refer to atmospheric models? Please make a consistent statement on the product usability with the one in your conclusions (lines 586-588).

**Lines 33-36:** The product landing page presents a different DOI  $\rightarrow$  https://doi.org/10.48670/moi-00170. Please check with the data publisher and display the correct one, together with the "How to Cite" instruction which is a best practice (see https://support.datacite.org/docs/landing-pages).

Line 67: "...the European Space Agency (ESA) Climate Change Initiative (CCI) SST, ..." add SST, or dataset, or product

**Lines 105-107**: "...Though model analyses by definition also assimilate observations, which could thus in principle include hourly SEVIRI data, in the present configuration they are not able to deal with such frequent updates (see section 2.2), and the approach presented here represents an effective way to improve the reconstruction of SST daily cycle from high-repetition satellite measurements..."

This phrase is not clear since you did not describe the MED-MFC system (physical component of the Med-MFC called Med-Currents) yet. Is the MedFS system assimilating or using (correction of the surface heat forcing) any satellite SST data? If yes, which one? Are SEVIRI data and MedFS model SST independent?

I suggest to insert in the intro some specification, i.e. resolution, accuracy (i.e. 0.76C when comparing SST to satellite L4 dataset, see page 5 of the QUID) about the **MedFS** (*Clementi et al.*

2021, see also https://medfs.cmcc.it/backend/public/medfs/short-description.html) as done for SEVIRI.

Lines 150-152: please use the citation and DOI, no need for the URL link. Please adopt a coherent approach for all your datasets references in the text. A suggestion would be to move the first paragraph in the introduction.

**Line 154**: please use MedFS instead of (MFS) as indicated at https://medfs.cmcc.it/backend/public/medfs/short-description.html. Please add also references at line 158:

- Clementi E., J. Pistoia, D. Delrosso, G. Mattia, C. Fratianni, A. Storto, S. Ciliberti, B. Lemieux, E. Fenu, S. Simoncelli, M. Drudi, A. Grandi, D. Padeletti, P. Di Pietro, N. Pinardi (2017). A 1/24 degree resolution Mediterranean analysis and forecast modeling system for the Copernicus Marine Environment Monitoring Service. Extended abstract to the 8th EuroGOOS Conference, Bergen.
- Clementi E., Oddo P., Drudi M., Pinardi N., Korres G. and Grandi A. (2017). Coupling hydrodynamic and wave models: first step and sensitivity experiments in the Mediterranean Sea. Ocean Dynamics. doi: https://doi.org/10.1007/s10236-017-1087-7.

Line 162: please include the right citation of the product, not the URL, and check the doi at the landing page https://doi.org/10.48670/moi-00172

**Table 1**: I wouldn't use "model" but either use the product identifier (or MedFS), since you are using a specific dataset.

Line 182: Same citation issue to be solved:

- https://resources.marine.copernicus.eu/productdetail/INSITU\_MED\_NRT\_OBSERVATIONS\_013\_035/INFORMATION there is a doi at this page without the citation instruction https://doi.org/10.48670/moi-00044
- https://resources.marine.copernicus.eu/product- detail/INSITU\_IBI\_NRT\_OBSERVATIONS\_013\_033/INFORMATION there is a doi at this page without the citation instruction https://doi.org/10.48670/moi-00043

Please be aware that it exists also this https://doi.org/10.13155/75807.

Line 199: The link you provided is not resolving to any landing page, please check and provide the right citation https://resources.marine.copernicus.eu/product-200 detail/SST\_GLO\_SST\_L4\_NRT\_OBSERVATIONS\_010\_001/INFORMATION The PUM link you provide afterwards reports a different product name, SST-GLO-SST-L4-NRT-OBSERVATIONS-010-014.

Lines 239-241: please check the English

Line 260: "*in the absence of* …" As mentioned earlier, this sentence seems misleading, since drifters data might be available, I suggest to erase it.

Line 264: which anomalies? The observation anomalies (*Fobs*-\_*Fb*\_)?

Line 274: I suggest to use the same nomenclature for MedFS SST or substitute "*model output*" with model SST (first layer model temperature).

Line 281: same comment as for line 162 above, please provide the DOI and citation instead of the URL.

**Line 290**: as suggested above, please avoid repetitions "*hourly MedFS seawater potential temperatures at 1.0182 meter (first level) characterized by 0.042° grid resolution.*

**Line 291**: What do you mean by regridding? Do you interpolate SEVIRI (0.05°) and MedFS SST onto the 1/16th grid before OI? How do you obtain the SST anomalies. Please specify.

Lines 317-325 are redundant if you improve the text of the paragraph answering the question above.

Line 350: I suggest to rephrase: "In order to evaluate assess the DOISST performance with respect to the MedFS SST analyses model and verify the correctness of the data blending, the same validation procedure has been applied to the modeled SST." In this case I would talk about verification since you verify that DOISST is improving with respect to its input dataset.

Line 373: I suggest "*DOISST and MedFS SST show similar but opposite behaviours*." I would also underline that the results are coherent with expectations, since MedFS represents the background field corrected by SEVIRI observations.

Figure 3, 4, 5 and captions, please add (a) (b) ...

Please consider to switch fig 3 and 4. I would prefer to see first the mean diurnal SST cycle as reconstructed by DOISST, MedFS and drifters, then to see the metrics.

Line 391: getting warmer  $\rightarrow$  warming

**Table 5**: I would put WINTER/SPRING/ in the first column and leave the months in the caption.

Line 424: I would take this out since you repeat it at line 431.

Line 437: 2021 or 2020?

**Line 439**: "*The grid resolution of OSTIA*..." This phrase is misleading, please clarify. Do you select the nearest grid point from each dataset? Do you use the MedFS regridded on the 1/16th grid? What about SEVIRI data. (this links to the issue above see the comment referred to Line 291)

Figure 9, caption and text: again, please use the specific name (i.e. MedFS) instead of the generic "model" or "model outputs".

**Data Availability section:** please use the right citation and DOI as asked above (Lines 33-36)

Line 532: please specify which model, consistently with satellite data, as suggested before.

**Line 536**: I suggest: "In an ideal case, all the DOISST input data and the validation dataset would be generated available and compared at the same depth."

Lines 545-546: "This product is also more accurate than the input model, which shows a mean bias of ~-0.1 K and RMSD 545 of ~0.47 K. A warm (positive) and cold (negative) bias characterizes the DOISST and the model, respectively, also during 546 seasons (Fig. 5). "I would strengthen here that DOISST is more accurate than MedFS SST as expected by the blending procedure, since it is used as background field, corrected by SEVIRI data. As I suggested before, I consider this part a verification of your blending procedure that successfully brings DOISST closer to the observed SST by drifters data.

Line 549: I suggest "...due to the vertical heat transfer heat process."

**Line 554**: "*The reduced bias could be ascribed to the fact that valid SEVIRI SST values are always interpolated in DOISST, while they are left unchanged in the original method*" This sentence is not clear to me, what do you mean by left unchanged? Did they impose the SEVIRI observed values? Please rephrase.

---

## Author Response (AR2)

**A New Operational Mediterranean Diurnal Optimally Interpolated SST Product within the Copernicus Marine Environment ₂ Monitoring Service**

Andrea Pisano, Daniele Ciani, Salvatore Marullo, Rosalia Santoleri, Bruno Buongiorno Nardelli

A new operational MEDiterranean Diurnal Optimally Interpolated Sea Surface Temperature (MED DOISST) product has been developed within the Copernicus Marine Environment Monitoring Service (CMEMS). MED DOISST is characterized by hourly mean maps (Level-4) of sub-skin SST at 1/16° horizontal resolution over the Mediterranean Sea from January 2019 to present. The sub-skin SST is the sea temperature at ~1 mm depth, which is subject to a large diurnal cycle.
The product is built by blending hourly SST data from SEVIRI and model analyses from the CMEMS MED-MFC system (Clementi et al. 2021) through optimal interpolation. The use of model SST as first-guess substitutes the adoption of climatologies or **previous analyses,** providing improved physically consistent estimates of hourly SSTs.

The manuscript after a first revision has been improved, answering the reviewers comments and it is accepted for further publication. However there are some issues, which I summarize hereafter, that I kindly ask you to take into consideration.

*We thank the reviewer for these comments that will furtherly improve the manuscript. Though not part of this review, we removed the expression 'Copernicus Marine Environment Monitoring Service', and then its acronym CMEMS, since it is today obsolete. We substituted it with Copernicus Marine Service.*

**Line 15**: I would specify which model dataset (as for SEVIRI) defining a name to keep over the manuscript --> some suggestions: MED-MFC (Clementi et al. 2021) or MedFS or Med-currents.

*We thank the reviewer for this comment. We now use 'MedFS' instead of 'model' in the whole manuscript.*

**Line 18**: *"...in the absence of any observation or in situ measurement"* This phrase is misleading since there are surface drifting buoys data that you use for validation. Why didn't you blend SEVIRI with in situ data? Could you please clarify?

*The background (or first-guess), as defined within the optimal interpolation (OI) theory, has to be a spatially- and temporally-complete field used to fill in the gaps generated by missing observations (satellite and/or in situ), and model data, by definition, satisfy this completeness criterion. In situ data can be used in OI as observations but not as background, since they do not satisfy the above completeness criterion. However, we agree that the sentence is misleading and has been removed.*

**Line 29**: This statement should be clarified, how would it improve the model predictability? Would you assimilate or use this product to correct the heat fluxes in other general ocean circulation models or do you refer to atmospheric models? Please make a consistent statement on the product usability with the one in your conclusions (lines 586-588).

*We rephrased this sentence as follows: "This product can contribute to improve the prediction capability of numerical models that assimilate or correct the heat fluxes starting from Level-4 SST data"*

**Lines 33-36:** The product landing page presents a different DOI ⮕ https://doi.org/10.48670/moi 00170. Please check with the data publisher and display the correct one, together with the "How to Cite" instruction which is a best practice (see https://support.datacite.org/docs/landing-pages).

*Corrected. The doi 'https://doi.org/10.48670/moi-00170' now replaces the previous one (https://doi.org/10.25423/CMCC/SST_MED_PHY_SUBSKIN_L4_NRT_010_036)*

**Line 67**: *"...the European Space Agency (ESA) Climate Change Initiative (CCI) SST,..."* add SST, or dataset, or product

*Corrected.*

**Lines 105-107**: *"...Though model analyses by definition also assimilate observations, which could thus in principle include hourly SEVIRI data, in the present configuration they are not able to deal with such frequent updates (see section 2.2), and the approach presented here represents an effective way to improve the reconstruction of SST daily cycle from high-repetition satellite measurements..."*
This phrase is not clear since you did not describe the MED-MFC system (physical component of the Med-Mfc called Med-Currents) yet. Is the MedFS system assimilating or using (correction of the surface heat forcing) any satellite SST data? If yes, which one? Are SEVIRI data and MedFS model SST independent?

I suggest to insert in the intro some specification, i.e. resolution, accuracy (i.e. 0.76C when comparing SST to satellite L4 dataset, see page 5 of the QUID) about the **MedFS** (*Clementi et al. 2021, see also https://medfs.cmcc.it/backend/public/medfs/short-description.html*) as done for SEVIRI.

*This paragraph has been rephrased, see lines 101-107 in the revised manuscript with track changes activated. We also modified Section 2.2, which introduces and describes MedFS, adding e.g. the accuracy of the first-layer temperature fields.*

**Lines 150-152**: please use the citation and DOI, no need for the URL link. Please adopt a coherent approach for all your datasets references in the text. A suggestion would be to move the first paragraph in the introduction.

*Corrected.*

**Line 154**: please use MedFS instead of (MFS) as indicated at *https://medfs.cmcc.it/backend/public/medfs/short-description.html*.

*Corrected.*

Please add also references at line 158:
- *Clementi E., J. Pistoia, D. Delrosso, G. Mattia, C. Fratianni, A. Storto, S. Ciliberti, B. Lemieux, E. Fenu, S. Simoncelli, M. Drudi, A. Grandi, D. Padeletti, P. Di Pietro, N. Pinardi (2017). A 1/24 degree resolution Mediterranean analysis and forecast modeling system for the Copernicus Marine Environment Monitoring Service. Extended abstract to the 8th EuroGOOS Conference, Bergen.*
- *Clementi E., Oddo P., Drudi M., Pinardi N., Korres G. and Grandi A. (2017). Coupling hydrodynamic and wave models: first step and sensitivity experiments in the Mediterranean Sea. Ocean Dynamics. doi: https://doi.org/10.1007/s10236-017-1087-7.*

*Added.*

**Line 162**: please include the right citation of the product, not the URL, and check the doi at the landing page https://doi.org/10.48670/moi-00172

*Corrected.*

**Table 1**: I wouldn't use "model" but either use the product identifier (or MedFS), since you are using a specific dataset.

*Corrected. 'Model' has been replaced by 'MedFS' in the whole manuscript.*

**Line 182**: Same citation issue to be solved:
- https://resources.marine.copernicus.eu/product detail/INSITU_MED_NRT_OBSERVATIONS_013_035/INFORMATIO N there is a doi at this page without the citation instruction https://doi.org/10.48670/moi-00044
- https://resources.marine.copernicus.eu/product detail/INSITU_IBI_NRT_OBSERVATIONS_013_033/INFORMATION there is a doi at this page without the citation instruction https://doi.org/10.48670/moi-00043 Please be aware that it exists also this https://doi.org/10.13155/75807.

*Solved.*

**Line 199**: The link you provided is not resolving to any landing page, please check and provide the right citation https://resources.marine.copernicus.eu/product-200 detail/SST_GLO_SST_L4_NRT_OBSERVATIONS_010_001/INFORMATION The PUM link you provide afterwards reports a different product name, SST-GLO SST-L4-NRT-OBSERVATIONS-010-014.

*Corrected. The DOI https://doi.org/10.48670/moi-00167 has been added.*

**Lines 239-241: please check the English**

*This sentence has been rephrased, see lines 236-240 in the revised manuscript.*

**Line 260**: " *in the absence of* ..." As mentioned earlier, this sentence seems misleading, since drifters data might be available, I suggest to erase it.

*This sentence was removed.*

**Line 264**: which anomalies? The observation anomalies (����− _�� _)?

*Corrected. Now reads: 'The observation anomalies'*

**Line 274**: I suggest to use the same nomenclature for MedFS SST or substitute "*model output*" with model SST (first layer model temperature).

*Corrected.*

**Line 281**: same comment as for line 162 above, please provide the DOI and citation instead of the URL.

Corrected.

**Line 290**: as suggested above, please avoid repetitions "*hourly MedFS seawater potential  temperatures at 1.0182 meter (first level) characterized by 0.042° grid resolution.*

*Corrected.*

**Line 291**: What do you mean by regridding? Do you interpolate SEVIRI (0.05°) and MedFS SST onto the 1/16$^{th}$ grid before OI? How do you obtain the SST anomalies. Please  specify.

*This sentence has been rephrased as follows: "Module M2 extracts and regrids (through*

*bilinear interpolation) both SEVIRI and MedFS SST data over the DOISST geographical domain at 1/16° grid resolution (see Table 2)''.*

*Concerning the second question, the definition of anomaly has been introduced in the background section. However, it is also repeated at the end of this section:*
- *Subtract hourly model SSTs from valid SSTs to produce SST anomalies;*

**Lines 317-325** are redundant if you improve the text of the paragraph answering the question above.

*If the reviewer agrees, we would prefer to leave this synthesis, that can be anyway useful as a recap. This synthesis scheme has also been rephrased to improve its clearness, see lines 311-327.*

**Line 350**: I suggest to rephrase: "*In order to  assess the DOISST performance with respect to  MedFS SST analyses  and verify the correctness of the data blending, the same  procedure has been applied to the modeled SST.*" In this case I would talk about verification since you verify that DOISST is improving with respect to its input dataset.

*This sentence was removed since it is now included at the beginning of Section 3.3. Indeed, also following another comment of the reviewer (Line 439), we made more homogeneous and compact the description of the validation framework.*

**Line 373**: I suggest "*DOISST and MedFS SST show similar but opposite behaviours.*" I would also underline that the results are coherent with expectations, since MedFS represents the background field corrected by SEVIRI observations.

*Here, since we are talking of mean bias, it would be preferable to leave the sentence as it is. However, we substituted 'model' with 'MedFS'.*

**Figure 3, 4, 5** and captions, please add (a) (b) …

*Corrected.*

**Please consider to switch fig 3 and 4. I would prefer to see first the mean diurnal SST cycle as reconstructed by DOISST, MedFS and drifters, then to see the metrics.**

*We agree with the reviewer that this switch would be of better impact. However, since this change would imply a restructuring of the whole section, we would prefer to leave the sequence as it is, if the reviewer agrees.*

**Line 391**: getting warmer ⬚ warming

*Corrected.*

**Table 5**: I would put WINTER/SPRING/ in the first column and leave the months in the caption.

*Corrected.*

**Line 424**: I would take this out since you repeat it at line 431.

*Removed.*

**Line 437**: 2021 or 2020?

*Corrected (2020).*

**Line 439**: "*The grid resolution of OSTIA…*" This phrase is misleading, please clarify. Do you select the nearest grid point from each dataset? Do you use the MedFS regridded on the $1/16^{th}$ grid? What about SEVIRI data. (this links to the issue above see the comment referred to Line 291).

*We do agree with the reviewer. We moved this sentence in Section 4.1, where the validation framework is introduced. We also clarified the space criterion (namely, nearest neighbor) and the grid resolution relative to DOISST, SEVIRI, MedFS and OSTIA diurnal.*

**Figure 9,** caption and text: again, please use the specific name (i.e. MedFS) instead of the generic "model" or "model outputs".

*Corrected.*

**Data Availability section:** please use the right citation and DOI as asked above (**Lines 33-36)**
*Corrected.*

**Line 532**: please specify which model, consistently with satellite data, as suggested before.

*Corrected.*

**Line 536**: I suggest: "*In an ideal case,  the DOISST input data and the validation dataset would be  available  at the same depth.*"

*Corrected.*

**Lines 545-546**: "*This product is also more accurate than the input model, which shows a mean bias of ~-0.1 K and RMSD 545 of ~0.47 K. A warm (positive) and cold (negative) bias characterizes the DOISST and the model, respectively, also during 546 seasons (Fig. 5).*" I would strengthen here that DOISST is more accurate than MedFS SST as expected by the blending procedure, since it is used as background field, corrected by SEVIRI data. As I suggested before, I consider this part a verification of your blending procedure that successfully brings DOISST closer to the observed SST by drifters data.

*We added this sentence: "These results also confirm the robustness of this blending algorithm that, even if based on model analyses used as first-guess, it successfully brings DOISST closer to the in situ measured SST than the MedFS estimates".*

**Line 549**: I suggest *"...due to the vertical heat transfer ."*

*Corrected.*

**Line 554**: "*The reduced bias could be ascribed to the fact that valid SEVIRI SST values are always interpolated in DOISST, while they are left unchanged in the original method*" This sentence is not clear to me, what do you mean by left unchanged? Did they impose the SEVIRI observed values? Please rephrase.

*We rephrased this sentence as follows: 'they are left unchanged (not interpolated, see section 3.3) in the original method '. This concept is actually introduced at the end of section 3.3.*

---

## Author Response (AR3)

**Dear Editor,**

Following your last comment, we have updated the reference Pisano et al. (2021) in order to be consistent with the corresponding DOI citation (https://doi.org/10.48670/MOI-00170).

Basically, Pisano et al. (2021) → CNR (2021), where the citation now reads:

CNR. (2021). Mediterranean Sea - High Resolution Diurnal Subskin Sea Surface Temperature Analysis [Data set]. MOi for Copernicus Marine Service. https://doi.org/10.48670/MOI-00170

Additionally, following the comments from the editorial team, we updated Tables 2-3-4 so that do not contain any more colored cells.

Best,

A. Pisano and Authors' Team.